# The 405 kyr and 2.4 Myr eccentricity components in Cenozoic carbon isotope records

Ilja J. Kocken[1], Margot J. Cramwinckel[1], Richard E. Zeebe[2], Jack J. Middelburg[1], and Appy Sluijs[1]

[1]Department of Earth Sciences, Faculty of Geosciences, Utrecht University, Princetonlaan 8a, 3584 CB, Utrecht, The Netherlands
[2]Department of Oceanography, University of Hawai'i at Mānoa, 1000 Pope Road, HI 96822, Honolulu, USA

**Correspondence:** Ilja J. Kocken (i.j.kocken@uu.nl)

**Abstract.** Cenozoic stable carbon ($\delta^{13}$C) and oxygen ($\delta^{18}$O) isotope ratios of deep-sea foraminiferal calcite co-vary with the 405 kyr eccentricity cycle, suggesting a link between orbital forcing, the climate system, and the carbon cycle. Variations in $\delta^{18}$O are partly forced by ice-volume changes that have mostly occurred since the Oligocene. The cyclic $\delta^{13}$C–$\delta^{18}$O co-variation is found in both ice-free and glaciated climate states, however. Consequently, there should be a mechanism that forces the $\delta^{13}$C cycles independently of ice-dynamics. In search of this mechanism, we simulate the response of several key components of the carbon cycle to orbital forcing in the Long-term Ocean-atmosphere-Sediment CArbon cycle Reservoir model (LOSCAR). We force the model by changing the burial of organic carbon in the ocean with various astronomical solutions and noise, and study the response of the main carbon cycle tracers. Consistent with previous work, the simulations reveal that low frequency oscillations in the forcing are preferentially amplified relative to higher frequencies. However, while oceanic $\delta^{13}$C mainly varies with a 405 kyr period in the model, the dynamics of dissolved inorganic carbon in the oceans and of atmospheric $CO_2$ are dominated by the 2.4 Myr cycle of eccentricity. This implies that the total ocean and atmosphere carbon inventory is strongly influenced by carbon cycle variability that exceeds the time scale of the 405 kyr period (such as silicate weathering). To test the applicability of the model results, we assemble a long (~22 Myr) $\delta^{13}$C and $\delta^{18}$O composite record spanning the Eocene to Miocene (34 to 12 Ma) and perform spectral analysis to assess the presence of the 2.4 Myr cycle. We find that, while the 2.4 Myr cycle appears to be overshadowed by long-term changes in the composite record, it is present as an amplitude modulator of the 405 and 100 kyr eccentricity cycles.

## 1 Introduction

During the Cenozoic (past 66 Myr) the Earth's climate system gradually shifted from an ice-free greenhouse world to an icehouse world with massive ice sheets at both poles (Zachos et al., 2001). Orbital forcing – quasi-cyclic changes in the climate system as a result of changes in the orbital configuration of the Earth (Milankovitch cyclicity) – caused superimposed variability in the climate-system drivers and carbon cycle on $10^4$ to $10^5$ year time scales. This has been shown in numerous palaeoclimate

proxy records, notably stable carbon isotope ($\delta^{13}$C) and oxygen isotope ($\delta^{18}$O) ratios of deep-sea foraminifera. Particularly distinctive is the imprint of the 405 kyr eccentricity term (Westerhold et al., 2011) on climate and the carbon cycle, under both ice-free (Paleocene–Eocene, Westerhold et al. 2011; Lauretano et al. 2015) and glaciated (Oligocene-Miocene, Pälike et al. 2006; Holbourn et al. 2015) climate regimes within the Cenozoic (Fig. 1). On this timescale, $\delta^{13}$C co-varies in phase with $\delta^{18}$O,

although the latter also responds strongly to the short eccentricity cycles (95 and 126 kyr). Similar to the shorter eccentricity cycles, the 405 kyr eccentricity cycle has a minor effect on total annual insolation (Laskar et al., 2004), but dominantly acts on the climate system as an amplitude modulator of precession, causing a greater seasonal contrast during eccentricity maxima. Changes in foraminiferal $\delta^{18}$O primarily represent changes in global ice volume and deep-sea temperatures. Changes in $\delta^{13}$C represent ocean-wide changes in the partitioning of carbon pools with different isotopic signatures among ocean, atmosphere,

and rock reservoirs. The co-variance of oxygen and carbon isotopes therefore implies a coupling of climate and the carbon cycle, where $\delta^{13}$C and $\delta^{18}$O minima correspond with eccentricity maxima, or periods of increased seasonal contrast. It is important to note, however, that orbital-scale variability is a function of both cyclic forcing and other, probably non-cyclic variability on different time-scales (e.g., from tectonics).

On time scales $>10^6$ yr, the $\delta^{13}$C of the global exogenic carbon pool is largely controlled by carbon burial on the continental

margins, where the isotopic composition of the buried carbon is mainly determined by the ratio between inorganic carbon and organic carbon ($C_{org}$) (Berner et al., 1983). Increases in $^{13}$C-depleted $C_{org}$ burial are caused by increased delivery of clays through river runoff, as $C_{org}$ is primarily buried in association with clay particles (Hedges and Keil, 1995). One might suspect that clay supply and burial increase during periods of increased seasonal contrast (eccentricity maxima), because this would optimize weathering, erosion, and transport. In this scenario, the resulting enhanced $C_{org}$ burial would lead to $\delta^{13}$C

maxima in the water column. However, we observe the opposite phase relation between eccentricity and $\delta^{13}$C in palaeorecords. Alternatively, clay formation may be optimized during phases of year-round relatively wet conditions in soils to facilitate silicate weathering and clay formation. If the system is not limited by transport rates, this scenario is more likely to occur during periods of low seasonal contrast (eccentricity minima), resulting in the observed phase relation between eccentricity and $\delta^{13}$C.

Various other forms of organic carbon sequestration may be the cause of the recorded phasing. For example, increased burial of $C_{org}$ may occur on land during eccentricity minima because constant precipitation above peat lands provides conditions for development of year-round soil anoxia (Kurtz et al., 2003; Zachos et al., 2010). The resulting $^{13}$C-enriched global exogenic carbon pool would be consistent with the phase relation we see in sediment records. This mechanism may be problematic, however, because it does not explain the consistent patterns throughout the Cenozoic, as terrestrial basins were unlikely stable

over tens of millions of years. Another factor that might influence the total mass and isotopic composition of inorganic carbon in the oceans involves submarine methane hydrate dynamics. Microbes in continental margin sediments produce biogenic methane from $C_{org}$. This methane, which has $\delta^{13}$C values of down to $-70\,‰$ VPDB, can be temporarily stored in methane hydrates (Dickens, 2003, 2011). The total mass of methane stored in hydrates is affected by deep sea temperature and could thus be controlled by orbital cycles. Assuming a steady sediment methane production, more biogenic methane is directly

mixed into the exogenic carbon pool under relatively warm climates (Dean et al., 2018). Under colder climates, more methane

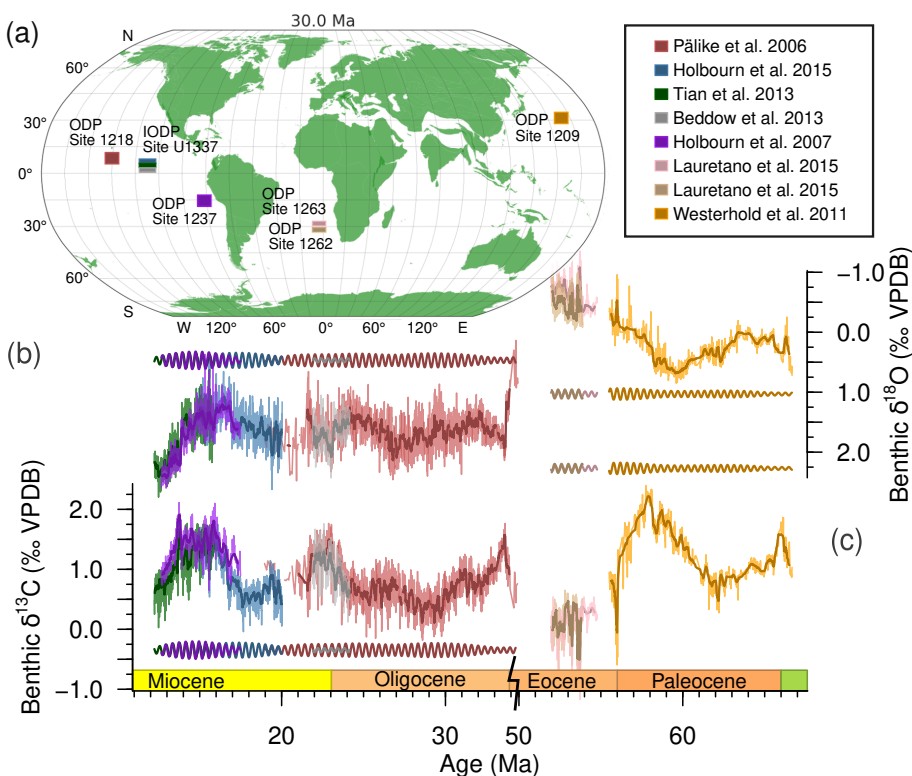

**Figure 1.** Long-term high resolution benthic foraminiferal stable carbon and oxygen isotope records show $405\,\text{kyr}$ cyclicity. (a) Palaeoceanographic reconstruction of $30\,\text{Ma}$ with approximate site locations. Map generated with GPlates, using the rotation frame and tectonic reconstruction of (Matthews et al., 2016). Data from references in the legend show the $405\,\text{kyr}$ periodicity in the $\delta^{18}\text{O}$ (b) and $\delta^{13}\text{C}$ (c) records throughout the Cenozoic at various sites in the Pacific. Darker lines going through the records represent 20-point moving averages while darker lines at arbitrary heights represent the $405\,\text{kyr}$ filtered records (at $2.47 \pm 0.15\,\text{Myr}^{-1}$). The Westerhold et al. (2011) record was converted to the GTS2012 (Gradstein et al., 2012) time scale by converting magnetic chron ages and placing the PETM onset at 56.0 Ma. Note the $\sim 16\,\text{Myr}$ x-axis break in the Eocene.

is trapped in hydrates, resulting in an exogenic carbon pool that is relatively $^{13}\text{C}$-enriched relative to the warm phase. This would also result in the co-occurrence of 405 and $100\,\text{kyr}$ eccentricity maxima and relatively negative $\delta^{13}\text{C}$ and $\delta^{18}\text{O}$ values. However, in Paleocene–Eocene sediment records, methane hydrate dynamics cannot explain the co-occurrence and cyclicity of $\delta^{13}\text{C}$ and $\delta^{18}\text{O}$, because they would result in a slope of $\Delta\delta^{13}\text{C} - \Delta\delta^{18}\text{O}$ that is very different from the one observed (Zeebe et al., 2017). They may play a role in the Oligocene and Miocene, (Pälike et al., 2006; Holbourn et al., 2015; Tian et al., 2013), but could also be related to changes in ice-volume (see below). Given the intermediate ocean-depth temperature response to orbital variability and seasonal contrast in a fully coupled climate model (Lunt et al., 2011), methane hydrate dynamics could be a plausible explanation for coupled $\delta^{13}\text{C}$ and $\delta^{18}\text{O}$ variability.

**Table 1.** Possible mechanisms that can cause $\delta^{13}$C–$\delta^{18}$O to co-occur, as a result of input eccentricity $\epsilon$ forcing maxima ($\uparrow$) and minima ($\downarrow$).

| Mechanism | $\epsilon$ | Response | Caveat/Prerequisite |
|---|---|---|---|
| Clay transport | $\uparrow$ | $\uparrow$ weathering $\Longrightarrow$ $\uparrow$ clay transport $\Longrightarrow$ $\uparrow$ C$_{org}$ burial $\Longrightarrow$ $\uparrow \delta^{13}$C | Does not match data |
| Clay production | $\downarrow$ | $\uparrow$ clay formation $\Longrightarrow$ $\uparrow$ C$_{org}$ burial $\Longrightarrow$ $\uparrow \delta^{13}$C | Weathering is not limiting |
| Peat-land soil storage | $\downarrow$ | year-round anoxia $\Longrightarrow$ $\uparrow$ C$_{org}$ sequestration in peat-land soils $\Longrightarrow$ $\uparrow$ remaining exogenic $\delta^{13}$C | Land-reservoir stable over millions of years |
| Methane hydrate reservoir | $\uparrow$ | $\uparrow$ seasonal contrast $\Longrightarrow$ $\uparrow$ intermediate water temperature $\Longrightarrow$ methane hydrate dissolution $\Longrightarrow$ $\downarrow \delta^{13}$C | Not in Paleocene–Eocene |
| Ice-sheet dynamics | $\uparrow$ | $\uparrow$ surface-air temperatures $\Longrightarrow$ $\downarrow$ in ice-volume sub-shelf melt, associated with $\downarrow \delta^{13}$C | Requires presence of ice-sheets |

Alternatively, ice-sheet dynamics have been proposed as the origin of the 405 kyr cycle for the past 35 Ma (de Boer et al., 2014). In a 3D ice sheet and climate model, increased insolation with higher surface-air temperature triggers a reduction in ice volume through sub-shelf melting, which can be linked to a decrease in $\delta^{13}$C-values through changes in the carbon cycle (de Boer et al., 2014). While the 405 kyr cycle in stable isotopes can be simulated adequately for the past 5 Myr (de Boer et al.,
2014), astronomical pacing of the carbon cycle in the early Paleogene – with the absence of permanent ice-sheets – means that this mechanism alone does not explain the link between the carbon cycle, astronomical forcing, and climate variability throughout the Cenozoic.

It is unknown which mechanism is responsible for the co-occurrence of the 405 and 100 kyr eccentricity cycles in climate records, but we do know that they must link astronomical forcing to changes in $\delta^{13}$C and $\delta^{18}$O. We decide to explore C$_{org}$ burial
in the oceans. In nature, continental margins provide the largest C$_{org}$ storage potential over astronomical time scales (Berner, 1982). C$_{org}$ burial is the product of sediment accumulation and organic carbon contents (corrected for porosity). C$_{org}$ burial is thus linearly related to sediment accumulation, which in turn is approximately linearly related to sediment delivery to the ocean in the absence of major changes in sea level (Berner, 1982). Other factors governing C$_{org}$ burial (e.g. oxygen) are secondary to sediment deposition (Hedges and Keil, 1995). Therefore, we focus our modelling efforts on linear astronomical forcing of
C$_{org}$ burial as a starting point. Since the model does not resolve shallow versus deep C$_{org}$ burial, we force the long-term balance between burial and oxidation of C$_{org}$ directly, without implying which particular mechanism is responsible.

Previous modelling studies have attempted to unravel the origin of the 405 kyr cyclic patterns found in Cenozoic $\delta^{13}$C and $\delta^{18}$O records. Pälike et al. (2006) hypothesized that the cycles are driven by variations in marine biosphere productivity in response to changes in insolation. Ma et al. (2011), however, focused their simulations on the effects of increased weathering
with eccentricity maxima, resulting in increased inorganic carbon input in the oceans, which, in combination with higher nutrients, leads to increased productivity and ultimately increased C$_{org}$ burial, coinciding with minima in global exogenic

$\delta^{13}$C. Paillard (2017), in a study on Pleistocene glacial-interglacial variability, argued that silicate weathering alone could not simultaneously account for the large $\delta^{13}$C changes and the small $p$CO$_2$ changes, while nutrient or ecology-induced changes in C$_{org}$ burial would not result in a change to these cycles during the onset of glaciations. Therefore Paillard (2017) focused on monsoon-driven changes in C$_{org}$ burial, which may be similar to our approach – as the monsoonal changes lead to changes in precipitation, which cause an increased riverine clay flux, resulting in increased C$_{org}$ burial. Zeebe et al. (2017) forced C$_{org}$ burial in marine sediments directly with various orbital forcing scenarios and noise, to understand a sediment record from the Paleocene and Eocene. These and previous modelling studies demonstrate how high-frequency spectral power in input forcing is dampened, resulting in increased power in the lower frequencies (Pälike et al., 2006; Ma et al., 2011; Paillard, 2017; Zeebe et al., 2017); The long residence time of carbon in the oceans (Broecker and Peng, 1982) effectively filters the input signal, resulting in a strongly muted response on short time scales, which allows only the lowest frequencies in the input forcing to become apparent in the record.

The question remains if this low-pass filtering also affects the much longer 2.4 Myr eccentricity cycle, for which indications have been found in geological records for the Eocene (Lourens et al., 2005), Paleocene (Westerhold et al., 2011), Oligocene (Pälike et al., 2006; Valero et al., 2014), and Miocene (Holbourn et al., 2007; Liebrand et al., 2016). The 2.4 Myr cycle is present in the astronomical solution as both an amplitude modulator (AM) of the 124 and 95 kyr eccentricity cycles as well as a true eccentricity cycle and is caused by orbit–orbit resonance between Mars and Earth ($g_4 - g_3$) (Laskar et al., 2011). This cycle is never unambiguous in palaeoclimate records, because long time series are required and the long period of this cycle overlaps with the time scale of slow tectonically-driven climate change ($>1$ Myr), and can therefore be obscured by the latter. The signal can also be lost when data are detrended prior to spectral analyses.

Here we aim to investigate the general role of eccentricity forcing on the carbon cycle across the Cenozoic, and specifically provide a model–data comparison with Oligocene and Miocene carbon isotope records. We linearly force C$_{org}$ burial in marine sediment with astronomical forcing in this modelling study, to assess the response of the carbon cycle.

## 2 Material and Methods

We investigate the response of the carbon cycle to variations in orbital parameters by conducting a modelling study with simple linear forcing of organic carbon burial, using the Long-term Ocean-atmosphere-Sediment CArbon cycle Reservoir model (LOSCAR, Zeebe, 2012). In the model, C$_{org}$ burial and weathering were forced with various aspects and combinations of the orbital solution by Laskar et al. (2004). We use spectral analysis techniques to study cyclic patterns in the model output. Finally, we compare modelling results to a long-term high-resolution data composite of published benthic foraminiferal isotope records from the Pacific Ocean.

### 2.1 LOSCAR model

The LOSCAR model simulates and tracks changes in the partitioning of carbon throughout the ocean basins, sediments and atmosphere over both short (centuries) and long (millions of years) time scales (Zeebe, 2012). The model consists of a number

of easily adjustable ocean boxes: in the set-up used here, it comprises an atmosphere, the Atlantic, Indian, and Pacific oceans, each with a surface-, intermediate-, and deep box, and one general high-latitude surface ocean box. All oceans are coupled to a prescribed number of sediment boxes (13 for each ocean here). A thermohaline circulation is imposed in the model by way of flow-rates between specific ocean boxes. Various tracers (e.g. dissolved inorganic carbon (DIC), $p\mathrm{CO_2}$, pH, and the $\delta^{13}\mathrm{C}$, as well as the carbonate compensation depth (CCD)) are tracked for the different ocean boxes in the model through time.

Carbon cycling in the model operates on various time scales that are relevant here, including volcanism and the long term weathering feed-backs, and the short-term gas exchange (similar to Berner et al., 1983). Long-term carbon cycling is implemented in the model as a flux of volcanic degassing and metamorphic processes ($F_{vc}^0$) and weathering of organic matter from the rock reservoir ($F_{OC}^{in}$), with atmospheric $\mathrm{CO_2}$ being consumed by carbonate weathering ($F_{cc}$, Reaction R1) and silicate weathering ($F_{si}$, Reaction R2).

$$CaCO_3 + CO_2 + H_2O \rightleftharpoons Ca^{2+} + 2\,HCO_3{}^{-} \tag{R1}$$

$$CaSiO_3 + 2\,CO_2 + H_2O \rightleftharpoons Ca^{2+} + SiO_2 + 2\,HCO_3{}^{-} \tag{R2}$$

These weathering reactions respond as a negative feedback to variations in the $\mathrm{CO_2}$ concentration as follows:

$$F_{cc} = F_{cc}^0 (p\mathrm{CO_2}/p\mathrm{CO_2^0})^{n_{cc}} \tag{1}$$

$$F_{si} = F_{si}^0 (p\mathrm{CO_2}/p\mathrm{CO_2^0})^{n_{si}} \tag{2}$$

Where $F_{cc}$ and $F_{si}$ are the carbonate and silicate weathering fluxes respectively, the superscript "0" refers to initial (steady-state) value of the weathering flux and $\mathrm{CO_2}$ respectively. The $n_{cc}$ and $n_{si}$ parameters control the strength of the weathering feedback. Steady state is reached when the silicate weathering flux balances the $\mathrm{CO_2}$ degassing flux from volcanism, i.e. $F_{si} = F_{vc}$. These are all slow processes, where the steady-state balance is restored after a perturbation on time scales in the order of $10^5$ to $10^6$ yr (Zeebe, 2012).

Carbon cycling also occurs on a much more rapid time scale in the oceans and atmosphere through the biological pump, air–sea exchange and remineralisation of carbon. In the original LOSCAR model (version 2.0.4), the biological pump is decoupled from the long carbon cycle described above, meaning that $\mathrm{C_{org}}$ that is produced in surface oceans is fully remineralised in the intermediate and deep ocean boxes (see Zeebe, 2012, for a full description). We note that Komar and Zeebe (2017) included a long-term, coupled $\mathrm{C_{org}}$-$\mathrm{O_2}$-$\mathrm{PO_4}$ feedback in LOSCAR. However, a simplified approach is used here (see below), since our simple forcing does not invoke additional controls on $\mathrm{C_{org}}$ burial, such as productivity.

## 2.2 Orbital forcing

We spun up the modern (i.e. post-Paleocene–Eocene) set-up of LOSCAR with atmospheric $CO_2$ concentrations of $450$ ppmv with default parameter values listed in Zeebe (2012) for $10$ Myr. Initial $CO_2$ concentrations of $450$ ppmv are roughly consistent with averaged late Paleogene to Neogene proxy estimates (e.g., Foster et al., 2012; Zhang et al., 2013). The equilibrium was then perturbed by linear scaling of the $C_{org}$ burial rate with the orbital forcing signal at each time step (Eq. 3), with higher burial occurring with weaker forcing ($F_{inp}$, some form of ETP, eccentricity, insolation, or clipped insolation).

$$Z = \frac{F_{inp} - \min F_{inp}}{\max F_{inp} - \min F_{inp}} \tag{3}$$

$$F_{orb} = 2(Z - \overline{Z}) \tag{4}$$

$$F_{gb} = F_{gb}^0 (1 - \alpha F_{orb}), \tag{5}$$

where $Z$ is the input forcing, $F_{inp}$, scaled between 0 and 1. The scaled orbital forcing, $F_{orb}$, is $Z$ minus its mean value, $\overline{Z}$, multiplied by 2 so that it is scaled between $-1$ and 1. $F_{gb}$ is the $C_{org}$ flux resulting from the initial steady-state $C_{org}$ flux, $F_{gb}^0$, being scaled with adjustable parameter $\alpha$ and $F_{orb}$. Thus, weaker input forcing is associated with increased burial, which in nature could be explained by – if weathering is not limiting (Sect. 1) – increased clay formation, delivery, and deposition, and associated burial of $C_{org}$.

To force these changes in $C_{org}$ burial, different combinations of the orbital parameters can be used. $F_{inp}$ was implemented as eccentricity or precession separately, or as artificial combinations of eccentricity, obliquity (tilt) and precession (together ETP), which is created by normalizing (subtracting the mean, then dividing by the standard deviation) and adding the ETP components, after scaling them with different factors. Forcing the model with individual components and ETP may seem rather artificial but allows us to study the impact of individual orbital components, and infers no mechanistic understanding of the underlying processes – such as the position of deep water formation. We also force with various insolation schemes with different latitudes and seasons.

To test the robustness of the carbon cycle response to changes in orbitally-forced $C_{org}$ burial, a parameterized amount of white noise (scaled between $-1$ and 1) was added to the burial rate. Noise was generated and applied when a time interval larger than $1$ kyr since the previous noise addition had passed. As additional sensitivity analyses, several (newly introduced) model parameters were changed to study their effect on model output. The atmospheric $CO_2$ concentration, the impact of the forcing on the $C_{org}$ burial ($\alpha$), the amount of noise, as well as the time step after which noise was added were adjusted to find how these affect model output. When using ETP as forcing, the relative make-up of the ETP curve, was varied, while for summer insolation different ways of artificial clipping were applied so as to simulate a non-linear response resulting in increased spectral power in the eccentricity components.

## 2.3 Data compilation

A high resolution (22 Myr long, 34 to 12 Ma, median sampling interval of 4 kyr) composite record of benthic foraminiferal $\delta^{13}$C and $\delta^{18}$O (Fig. 1) was compiled using data from Ocean Drilling Program (ODP) Leg 199 Site 1218 (8°53.378′N, 135°22.00′W, 4.8 km water depth, Pälike et al. (2006)) and Integrated ODP (IODP) Site U1337 (3°50.009′N, 123°12.352′W, 4.2 km water depth, Tian et al. (2013); Holbourn et al. (2015)), located in the equatorial East Pacific (Fig. 1 (a)). We use records from the Pacific, because it represents the largest ocean reservoir by volume, and compare these to output from LOSCAR's Pacific box. Initial age models for the records were established from $^{40}$Ar/$^{39}$Ar isotopes, planktonic foraminiferal-, nannofossil-, radiolarian-, and/or magnetostratigraphic datum events, and known ages of changes in wt % CaCO$_3$ contents. The age models of these records are based on similar dating and tuning strategies and we therefore assume that they are compatible for the purposes of this long-term study. Initial age models were subsequently improved through astronomical tuning to the orbital solution Laskar et al. (2004). Pälike et al. (2006) tuned their record to an Eccentricity Tilt Precession (ETP) model (with ratio 1:0.5:−0.4) starting with the 405 kyr cycle and then moving on to the shorter astronomical frequencies, while Holbourn et al. (2015) tuned their $\delta^{18}$O minima to ETP (1:1:0.2) maxima and Tian et al. (2013) tuned $\delta^{18}$O minima to obliquity maxima (which represents the same phase relation).

Areas of overlap in the various records comprising the data composite were checked for consistency in phase relationship, absolute value (Supp. Sect. 3) and amplitude variation. The $\delta^{13}$C and $\delta^{18}$O composite data from Pälike et al. (2006) were limited to the higher-resolution range between 20 to 34 Ma to keep the resolution comparable to the other records. Due to minor inconsistencies in the two age models for site U1337, data from Tian et al. (2013) were excluded in the overlapping depth region because of the lower resolution and shorter record length compared to the Holbourn et al. (2015) record.

## 2.4 Spectral analysis

All analyses were performed using the R programming language (R Core Team, 2018), including various functions of the Astrochron package (Meyers, 2014), unless indicated differently. To identify periodicity in both the model output and the data composite, the multitaper method (MTM) by Thomson (1982) was applied. First, data and model output were linearly re-sampled to the median sampling interval of the data composite (4 kyr). For the MTM-analysis, two tapers were used and the data or model output were zero-padded with a length of five times the input length. The data composite was also filtered for periods larger than 3.1 Myr to remove longer-scale climate trends that likely have causes other than orbital forcing, to allow for easier model–data comparison of records and spectra. The value of 3.1 Myr was chosen based on a small trough in spectral output of the raw data composite around this value (Sect. 3). This was achieved by applying a low-pass filter (for periods below 3.1 Myr), subtracting this signal from the original composite and re-adding the mean of the low-passed signal. Cross-correlation was performed between model-output and input forcing, to estimate phase-lags and provide insight into the dominant mechanisms determining model response to orbital forcing of C$_{org}$-burial. This was achieved with the Blackman-Tukey method, using a Bartlett window, using 30 % of the series, with a time-step of $1 \times 10^{-5}$ cycles/kyr using Analyseries 2.0.8 (Paillard et al., 1996). Amplitude modulation of shorter cycles was investigated by filtering for the short eccentricity

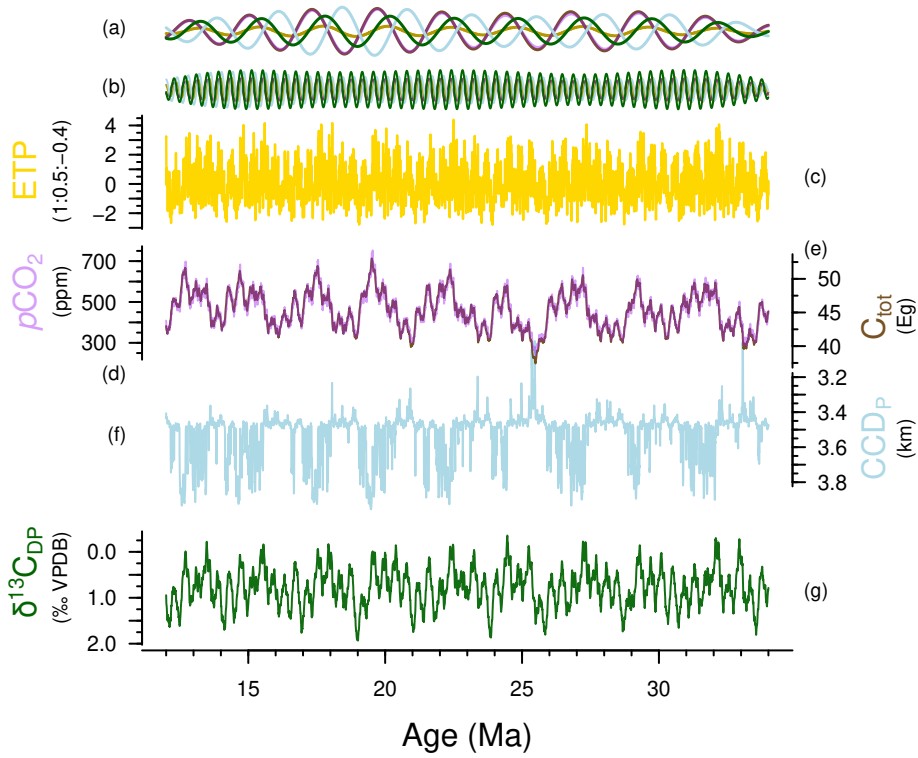

**Figure 2.** LOSCAR model output time series for orbitally-forced carbon burial with orbital forcing strength $\alpha = 0.5$ and noise level 0.2. (a) 405 kyr (frequency of $2.46 \pm 0.15\,\mathrm{Myr^{-1}}$) and (b) 2.4 Myr ($0.42 \pm 0.15\,\mathrm{Myr^{-1}}$) bandpass filters are shown (same colours, arbitrary y-axis). (c) The ETP (1:0.5:$-0.4$) construct used as orbital forcing in this scenario (gold). (d) Atmospheric $CO_2$ (purple, parts per million volume, ppm) and (e) the total exogenic carbon inventory (brown, Eg ($10^{18}$ g)). (f) Pacific carbonate compensation depth (CCD) (blue, km, note the cutoff y-axis due to large outliers, explained in Supp. Sect. 1) (g) Deep Pacific $\delta^{13}$C (green, ‰ Vienna Pee Dee Belumnite (VPDB), reversed axis). Deep Pacific temperature ($^\circ$C), and Dissolved Inorganic Carbon (DIC) of the Deep Pacific model output are omitted from this plot, because their pattern mirrors that of $C_{tot}$ and $pCO_2$ (Supp. Fig. 10).

cycles ($0.010 \pm 0.003\,\mathrm{kyr^{-1}}$), performing a Hilbert transform which connects all the peaks in the record, and applying the MTM on the result (adapted from Zeebe et al., 2017). To evaluate how the frequency distribution and amplitude modulation changes through time in the composite record, evolutive multitaper harmonics analysis (EHA) was performed on the 1 Myr negative filtered composite of $\delta^{13}$C using a time step of 50 kyr and a window size of 0.7 Myr (similar to Pälike et al., 2006;
5   Liebrand et al., 2016).

## 3   Results and Discussion

### 3.1   Model output

All tracers in the model output show cyclic alternations as a result of implementing astronomical forcing of $C_{org}$ burial in the model (Fig. 2 and Supp. Fig. 10 for all model output). Model tracers do not show much inter-basin variability, thus we will consider the deep Pacific box (DP) to assess the generalized oceanic behaviour of tracers. As expected, the total exogenic carbon inventory $C_{tot}$ in the ocean basins closely matches changes in atmospheric $CO_2$, dissolved inorganic carbon (DIC) and temperature. While the carbonate compensation depth (CCD) of the DP shows cyclic extreme outliers due to model design (Supp. Sect. 1), the underlying pattern of change is also similar, with high temperatures, high $CO_2$ concentrations, and high $C_{tot}$ co-occurring with a deepening of the CCD for the $405\,kyr$ and $2.4\,Myr$ cycles. DP $\delta^{13}C$ co-varies approximately in anti-phase (i.e. $^{13}C$-depleted DIC co-occurring with high $C_{tot}$) with these tracers but shows more higher-frequency fluctuations.

The MTM-analysis of model output shows significant spectral power at orbital periods (Fig. 3). Most notably, decreased spectral power occurs in the higher frequencies (precession, obliquity) in the forcing signal, while the lower frequencies (long-term eccentricity cycles), that are sometimes only marginally present in the input forcing gain spectral power. This low-pass filtering of spectral power, occurs for our main scenario of $1:0.5:-0.4$ ETP, but also for various other combinations of ETP (e.g. $1:0.5:-0.4$, $0.1:0:0.1$, $1:1:1$, etc.), eccentricity alone, and unclipped/clipped $65°N$ and $30°N$ summer insolation (Supp. Sect. 4). Results are qualitatively similar for different contributions of E, T, and P in the ETP curve as forcing. Furthermore, the results are largely unaffected by the introduction of relatively strong white noise, that is transformed into red noise, that affects $C_{org}$ burial by up to the same levels as the astronomical forcing, and different initial modelling conditions. Together, these sensitivity studies indicate that the model results in terms of spectral response are very robust (Supp. Sect. 4).

By comparing the dominant frequencies in our output to the characteristic frequencies of orbital solution from Laskar et al. (2004), the corresponding orbit–orbit resonances between the planets were found (Supp. Table 1). Surprisingly, the $2.4\,Myr$ cycle in carbon and oxygen isotopes is strongly present in the model output (including DIC, atmospheric $CO_2$, deep Pacific temperature and $C_{tot}$). The other strong eccentricity cycles consist of a $1\,Myr$ cycle that is related to a Mercury–Jupiter ($g_1 - g_5$) resonance, and the $405\,kyr$ cycle that is related to Venus–Jupiter ($g_2 - g_5$).

Interestingly, the low-pass filtering occurs differently for the different tracers. The DIC, atmospheric $CO_2$, $C_{tot}$ and temperature show very high relative dominance of the long eccentricity cycles ($2.4$, $1$ and $0.7\,Myr$), while $\delta^{13}C$ – and to a lesser extent CCD – show more spectral power in the shorter eccentricity range ($405$, $127$ and $95\,kyr$) (Fig. 3). To investigate why spectral power is filtered differently for these tracers, residence times were calculated for the key carbon cycle feedbacks in the model by dividing the reservoir sizes by the respective fluxes (See Table 2). Silicate weathering, with a residence time of $655\,kyr$, is the slowest feedback system in our model. Carbonate weathering operates on timescales of the same order of magnitude ($280\,kyr$), while oceanic carbonate compensation is an order of magnitude faster ($10\,kyr$). This means that silicate weathering most strongly low-pass filters the spectral power, resulting in a dominant $2.4\,Myr$ eccentricity cycle.

**Table 2.** Calculated residence times of the main carbon cycle processes.

| Process | Reservoir / Flux | Residence time |
|---|---|---|
| SiO$_2$ weathering | $TCO_2/F_{si}$ | 655 kyr |
| CaCO$_3$ weathering | $TCO_2/F_{cc}$ | 280 kyr |
| Carbonate compensation | $TCO_3{}^{2-}/F_{cc}$ | 10 kyr |

The $\delta^{13}$C gradients of carbon in the oceans are modulated by changes in the ($^{13}$C-depleted) C$_{\text{org}}$ pump. The biological pump operates much more rapidly than silicate- and carbonate weathering, but remains constant in the model. Therefore, changes in the $\delta^{13}$C gradients are dominated by the total C$_{\text{org}}$ inventory, on which we directly impose our forcing.

### 3.1.1 Phase relations of model tracers

Cross-spectral analysis between the key carbon cycle tracers in the model output and the ETP-forcing reveals different phasing for the various tracers, as well as for the different time-scales under consideration (Fig. 4). A phase of $0°$ implies that the tracer is changing exactly in phase with the ETP-forcing, while at $180°$ or $-180°$ they are in anti-phase. Positive (negative) values indicate that the variable is lagging (leading) with respect to the ETP-forcing.

Broadly, we observe that phasing remains rather constant for each tracer for periods $<405$ kyr, while absolute lag-time
decreases as a function of frequency (Supp. Fig. 7). The atmospheric CO$_2$ concentration and ocean temperature show the smallest lags – or fastest response – compared to the oceanic tracers. This is because the much shorter residence time of CO$_2$ in the atmosphere (in the order of several years) – due to its relatively small reservoir size and fast cycling – causes atmospheric CO$_2$ concentrations to respond rapidly to changes in sea surface carbon. The fast temperature response occurs because ocean temperature change propagates at a higher rate than changes in ocean chemistry. Atmospheric CO$_2$ and ocean temperature
therefore appear to be in quasi-steady state with respect to the other tracers in the model. Furthermore, cycling in the deep (D) and intermediate (I) Pacific (P) boxes is slower than the low-latitude surface (L) boxes for DIC. This is also a result of relative reservoir size and air-sea exchange. The CCD has a negative phase for the higher frequencies (precession, obliquity, and $100$ kyr eccentricity), indicating anti-phasing relative to the other tracers. With ETP-maxima, the CCD shoals at these shorter time-scales due to relatively rapid carbonate compensation as a short-term response to increased atmospheric CO$_2$ (Table 2).
On longer time-scales ($>100$ kyr), however, the CCD shifts to in-phase cycling – deepening of the CCD with ETP-maxima – as a result of increased weathering and increased riverine input of DIC and TA, which increases the oceanic DIC inventory but also restores alkalinity to the ocean.

For the phase-analysis, $\delta^{13}$C minima were compared to ETP maxima (i.e. $-\delta^{13}$C was compared to ETP), while the other tracers (CCD, C$_{\text{tot}}$, CO$_2$) were not adjusted. The high-frequency range shows similar lags to the DIC and C$_{\text{tot}}$. Surprisingly,
the DP box shows a slightly faster response than the LP and IP boxes, in fact more akin to the atmospheric reservoir for the

precession frequencies. With decreasing frequency (i.e. the $405\,\mathrm{kyr}$ to $2.4\,\mathrm{Myr}$ eccentricity components), this relation between the deep and shallower boxes switches and the $-\delta^{13}\mathrm{C}$ starts to lag more and more, reaching an overshoot of more than half a period for the $2.4\,\mathrm{Myr}$ cycle in the DP box, causing the $2.4\,\mathrm{Myr}$-cycle in $-\delta^{13}\mathrm{C}$ to appear as leading the signal in the ETP-forcing (Supp. Fig. 2). This can be demonstrated to be an overshoot response (Supp. Sect. 5). It is also interesting to note that
$\delta^{13}\mathrm{C}$ leads over DIC. This is because the DIC-response is determined by ocean chemistry and thus weathering, whereas the $\delta^{13}\mathrm{C}$-signal is mostly influenced by residence time and $\mathrm{C_{org}}$ cycling.

### 3.2 Data composite

The composite record consists of a carbon isotope record of relatively high resolution (median sampling interval of $\sim4\,\mathrm{kyr}$), spanning the Oligocene and most of the Miocene ($34$ to $12\,\mathrm{Ma}$). However, there is a lower resolution stretch of data between
10 $20.0$ and $21.5\,\mathrm{Ma}$ (Fig. 1). The highest and lowest (Nyquist and Rayleigh) detectable frequencies are thus $F_N = 1/(2\Delta t) = 0.125\,\mathrm{kyr^{-1}}$ and $F_R = 1/(N\Delta t) = 0.046\,\mathrm{Myr^{-1}}$ respectively.

   The dominant frequencies in the data composite correspond to the main orbital periods of eccentricity ($405$, $124$ and $95\,\mathrm{kyr}$), obliquity ($41\,\mathrm{kyr}$), and precession ($23$, $22$ and $19\,\mathrm{kyr}$) (Fig. 5), similar to the results of the individual records (Pälike et al., 2006; Tian et al., 2013; Holbourn et al., 2015). Despite detrending, very high spectral power remains in the very low frequency
range ($<0.5\,\mathrm{Myr^{-1}}$) (Fig. 5 (c)). This could be caused by non-periodic long-term climatic variations, or possibly by bundles of $2.4\,\mathrm{Myr}$ cycles (Boulila et al., 2012).

   The dominant $2.4\,\mathrm{Myr}$ cyclicity observed in our model output seems absent in the $\delta^{13}\mathrm{C}$ composite record when performing MTM spectral analysis. The EHA records a strong influence of the $405$, $127$ and $95\,\mathrm{kyr}$ cycles throughout the composite record, which vary in intensity over $2.4\,\mathrm{Myr}$ intervals (Fig. 6). Furthermore, the envelope of the $100$ and $405\,\mathrm{kyr}$ filtered record also
changes at $2.4\,\mathrm{Myr}$ intervals (Supp. Fig. 8). This indicates that the $2.4\,\mathrm{Myr}$ cycle acts purely as an amplitude modulator (AM) on the other eccentricity cycles in the composite record, and not as a true cycle.

   As a true cycle, the $2.4\,\mathrm{Myr}$ signal may be overshadowed by spectral power in the lower-frequencies (Fig. 5). If these are filtered out (periods $<3.1\,\mathrm{Myr}$, red shaded region in Fig. 5 (c)), the AR-1 fit and $95\,\%$ confidence levels are lowered on the low-frequency range ($<20\,\mathrm{Myr^{-1}}$), but spectra are unaffected. The relatively high spectral peak in the $2.4\,\mathrm{Myr}$ range of the
25 low-pass filtered data composite is therefore interpreted to be a combined result of a weak (overshadowed) signal in the data, and an artefact produced by cutting off (low-pass filtering) the even lower frequencies, leaving only this frequency range in the spectral outcome. The obliquity and precession components also change through time, but contribute less power to the EHA-spectrum. Qualitatively, a weak $1\,\mathrm{Myr}$ bundling can be inferred for these frequencies, especially for the older range ($25$ to $34\,\mathrm{Ma}$). This could be related to the $\sim1\,\mathrm{Myr}$ obliquity cycle.

### 30 3.3 Model–data comparison

The model generates quite comparable time series to the high-passed data composite in terms of amplitude, phasing, and associated spectra. The distribution of spectral power in astronomical components is similar, and can be matched to fit the data composite more closely by adjusting the strength of the orbital forcing, the make-up of the ETP, or the noise level (Supp.

Sect. 4). However, while the 2.4 Myr eccentricity cycle is predominantly present as a true cycle in the model output, the composite record instead contains a strong presence of this cycle as amplitude modulator of the shorter eccentricity cycles. Upon further inspection, we also find ~2.4 Myr bundling of the short eccentricity cycles in the EHA of model output (Supp. Fig. 9. This is probably the result of the eccentricity component in the ETP, as this bundling is absent when forcing with, e.g.

65°N summer insolation. The changes in temperature ($\sim 3\,°C$) that were calculated from model output $CO_2$ (Supp. Fig. 10) are larger than the temperature estimates based on the $\delta^{18}O$ composite data (which would be $\sim 1\,°C$). This is because the carbon cycle model (not a climate model) relates temperature to $CO_2$ directly through an input climate sensitivity of $5\,°C$ per $CO_2$ doubling, whereas in reality temperature is a function of a number of variables, including but not limited to $p\,CO_2$. For example, insolation directly affects the climate system components (e.g. atmosphere, etc.), which in turn affect temperature. These

insolation-driven changes to the climate system are not part of the temperature-response to $p\,CO_2$, however. This behaviour cannot be captured by the carbon cycle model. In addition, this climate sensitivity strongly depends on the strengths of various feedbacks at various time scales, which likely also vary through time (e.g., Rohling et al., 2012; von der Heydt et al., 2016). The lack of a strong 2.4 Myr cycle in $\delta^{18}O$ records (and its associated temperature) therefore does not necessarily imply that the 2.4 Myr cycle cannot be dominant in $p\,CO_2$.

Unfortunately, even the available $p\,CO_2$ records for the Pliocene–Pleistocene (e.g., Bartoli et al., 2011; Seki et al., 2010) are of insufficient resolution (~200 kyr, potential for aliasing) and duration (~5 Myr, at best 2 cycles) to test for a 2.4 Myr cycle. Furthermore, potentially cyclic changes could be overshadowed due to proxy uncertainty. New, long $p\,CO_2$ proxy records are required to establish a possible 2.4 Myr cyclicity. Since the resulting model $p\,CO_2$ can be adjusted by changing input parameters of forcing, noise, etc. (Supp. Sect. 4), comparison in terms of absolute values is not meaningful.

On long time-scales, the model's CCD output can be compared to reconstructions of the Pacific CCD based on $CaCO_3$ accumulation rates (CAR) from various sites (Pälike et al., 2012). These records indicate a deep and stable CCD during the Oligocene–Miocene up to about 18.5 Ma, but our modelled CCD depth is highly variable on 405 kyr and 2.4 Myr time-scales – ranging from ~10 to 100 m if we exclude model-setup-induced outliers (Supp. Sect. 1). The CCD reconstructions for the Eocene, however, are highly variable on these time-scales (changing by up to ~500 m) and potentially show indications for

long-term eccentricity-forced cycles. We performed spectral analyses on the CAR for the different sites as well as the inferred Pacific CCD (Supp. Figs. 1 and 2). In these records, long-term trends and step changes, are controlled by long-term non-cyclic forcings such as volcanism, sea level or long-term changes in the climate (Pälike et al., 2012). Therefore, we remove the long-term trend using a Loess fit prior to spectral analysis. These analyses provide some evidence for 2.4 Myr cyclicity in the longest continuous CAR records for Site 1333 and Site 1334 (Supp. Fig. 1) as well as for the global CCD reconstruction (Supp. Fig. 2).

The evidence is far from strong, however, because also these records are of low resolution – insufficient to resolve the short (100 and 405 kyr) eccentricity cycles and certainly precession. Furthermore, there are indications that the 2.4 Myr eccentricity cycle that is present in the astronomical solution is expressed as a 1.2 Myr AM of the short eccentricity cycle during the Eocene, due to either a resonance with, or a non-linear response of the residence time of gas hydrates (Galeotti et al., 2010), further complicating the matter. The fact that we see some evidence for the 2.4 Myr cycle, given these caveats, however, is noteworthy.

The phasing of the model and data records differ in terms of absolute lag (kyr): the modelled $\delta^{13}$C response is overall slower compared to data records. For example, the lag of $\delta^{13}$C to the $405\,$kyr eccentricity cycle is ~$190\,$kyr, which is much higher than the $40$ to $60\,$kyr estimates for the Paleocene (Westerhold et al., 2011) and $20$ to $30\,$kyr during the Oligocene (Pälike et al., 2006). Of course the lag times of data records are calculated after the tuning of a record, and are therefore likely underestimated.

However, the large difference between modelled lag-times and lag-times calculated from tuned records may indicate a lack of understanding regarding the underlying mechanisms.

With simple linear $C_{org}$ burial forcing using an artificial combination of eccentricity, precession and tilt, we are able to generate a $\delta^{13}$C profile for the Eocene–Miocene that is very comparable to a long term high-passed data composite (Fig. 5) in both the time (Fig. 2) and frequency domains (Fig. 3). This indicates that cyclic burial of $C_{org}$ should be considered as a

potential candidate driving $\delta^{13}$C and $\delta^{18}$O variability on eccentricity time scales.

## 3.4 Outlook

Looking forward, we could utilise phase-lag results of modelling studies to further constrain the lag between astronomical forcing and sedimentary records, allowing for an even more precise absolute astronomical tuning. Currently, most studies assume no phase-lag between astronomical forcing and the palaeorecord due to lack of absolute ages relative to the astronomical

solution.

Future work could focus on evaluating the different magnitudes of the spectral power changes towards lower frequencies between different carbon cycle variables in the palaeorecord, as modelled $\delta^{13}$C of $C_{org}$ shows a less-pronounced dampening of high-frequencies compared to other model tracers such as the CCD, total exogenic carbon, temperature or atmospheric $CO_2$. Ideally this would require high-resolution records (at least $5\times$ the Nyquist frequency for the period of interest, i.e. every $2.3\,$kyr

in order to resolve precession) long-term (ideally >5 $2.4\,$Myr cycles, $12\,$Myr) records for various climate variables, and could prove quite a challenge.

## 4  Conclusions

The carbon cycle box model simulations presented here reveal that simple orbital forcing of organic carbon burial in the ocean can reproduce the dominant $405\,$kyr cycle that is observed in Cenozoic carbon isotope records. The good correspondence be-

tween modelled carbon isotopes and the palaeorecord confirms orbitally forced burial of organic carbon as a possible candidate for causing the observed $405\,$kyr cyclicity in $\delta^{13}$C.

The link between astronomical forcing, carbon cycle dynamics, and the climate system that may give rise to this consistent eccentricity-forced periodicity is still poorly understood. Here we show that astronomical forcing of $C_{org}$ burial in the marine realm could be a mechanism that explains this link in both a glaciated and ice-free world. As $C_{org}$ burial is primarily controlled

by sediment accumulation (Berner, 1982; Hedges and Keil, 1995; Berner, 2006), the direct top-down control for $C_{org}$ burial would be river runoff, which corresponds to seasonal precipitation patterns (see also Paillard, 2017; Zeebe et al., 2017). Although the exact mechanistic link between eccentricity forcing and sediment deposition remain elusive, we demonstrate that

marine $C_{org}$ burial provides an alternative to other proposed mechanisms that link the carbon cycle to astronomical forcing, such as carbon storage in peat-land soils (Kurtz et al., 2003), methane hydrate reservoirs, or ice-sheet dynamics (de Boer et al., 2014).

Importantly, this modelling study implies a significant role for the $2.4\,$Myr eccentricity cycle in modulating carbon cycle dynamics, especially in DIC and atmospheric $CO_2$. The $2.4\,$Myr eccentricity cycle is also present in the data composite, but acts dominantly as an amplitude modulator of the $127$ and $95\,$kyr eccentricity cycles, and to a lesser extent as a true cycle, possibly due to overshadowing by slower cycles. The hypothesis that the Cenozoic $p$CO$_2$ spectrum was dominated by the $2.4\,$Myr eccentricity could be evaluated using long, high-resolution proxy records of $p$CO$_2$ or other carbonate chemistry parameters.

*Code and data availability.* The LOSCAR model source code is available upon request to loscar.model@gmail.com. The implementation of astronomical forcing and noise addition to the model are available as a patch file on the corresponding author's GitHub (https://github.com/japhir/cp-2018-42).

The data that were used to create the data composite are available from the respective references and their associated entries on the Pangaea database (https://pangaea.de). The R-code that was used to assemble the composite is available on the above-mentioned GitHub repository.

*Author contributions.* Modelling work was conducted by MJC and IJK. Data compilation was generated by IJK. All authors provided feedback on the manuscript.

*Competing interests.* The authors declare that they have no conflict of interest.

*Acknowledgements.* IJK thanks the kfHein fund for financial support. MJC thanks the Molengraaff Fund for financial support. REZ was supported by U.S. NSF grants OCE12-20615 and OCE16-58023. This work used data generated on samples provided by the International Ocean Discovery Program (IODP) and was carried out under the program of the Netherlands Earth System Science Centre (NESSC). We also thank Frits Hilgen for extensive discussions and two anonymous reviewers for their input on the manuscript.

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

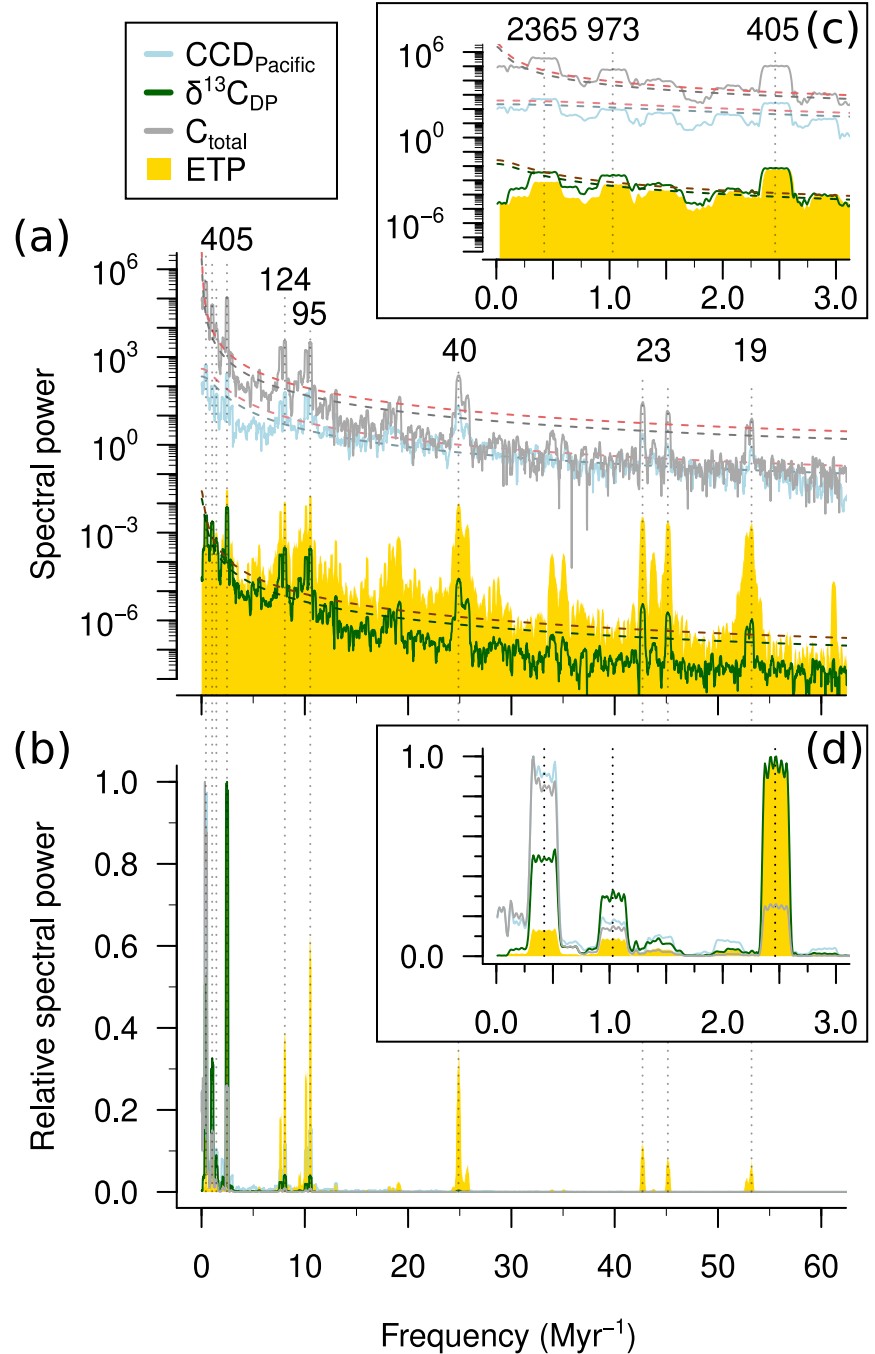

**Figure 3.** MTM spectral analysis of LOSCAR model output with orbital forcing strength $\alpha = 0.5$ and noise level 0.2. Shown are the model tracers and the ETP forcing used on absolute (a) (log axis) and normalized (b) (linear axis) spectral power. Relative spectral power was calculated by dividing each value by their maximum value. Insets (c) and (d) are the same as (a) and (b), but zoomed in on the frequency axis. Periods of interest ($kyr^{-1}$) are labelled and indicated with dashed lines.

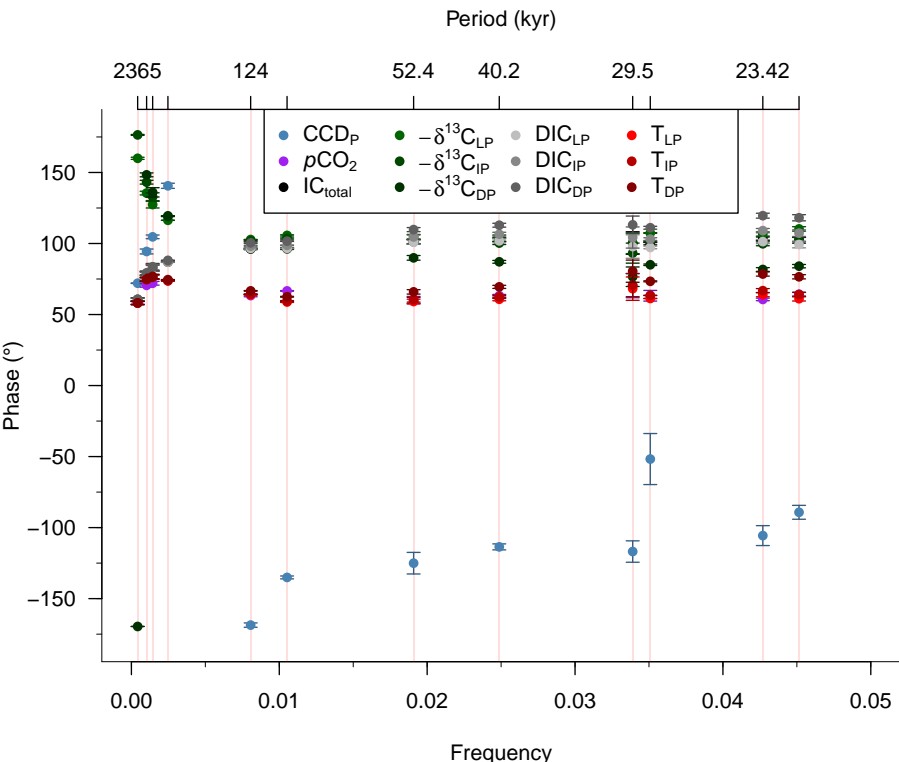

**Figure 4.** Phase lag (°, y-axis) of the various modelled carbon cycle tracers (see legend, L stands for low-latitude surface, I for intermediate, and D for deep ocean box of the Pacific (P)) with respect to the ETP forcing (of strength $0.5$ and noise $0.2$) as a function of frequency (cycles/kyr, bottom x-axis) and period (kyr, top x-axis). Note minor break in x-axes to better show the low-frequency range. Error bars represent $2\sigma$ of the variation of the phase over the specified frequency intervals (red shaded regions). Note that $-\delta^{13}$C is shown, such that a phase-lag (positive values) represents a lag of $\delta^{13}$C-minima with respect to ETP-maxima.

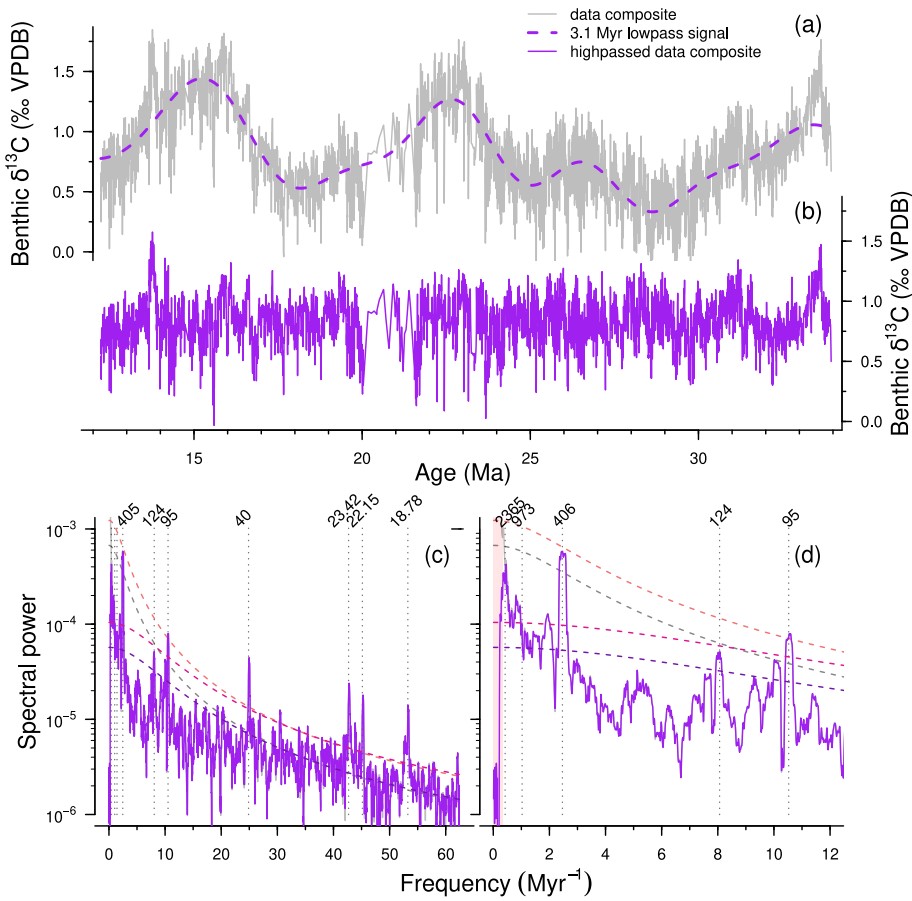

**Figure 5.** (a) $\delta^{13}$C composite record (grey line) with a low-pass filter for frequencies below 3.1 Myr (dashed purple line). (b) The adjusted composite (purple) in the time domain. (c) The same records in the frequency domain. Dashed smooth lines represent the AR-1 fits for the $\delta^{13}$C and detrended $\delta^{13}$C records (grey and purple respectively) with associated 95 % confidence intervals (light and dark red respectively). Main orbital periods (kyr) are labelled and marked with dotted lines. (d) A zoomed-in version of (c) on the frequency axis, with the red shaded region indicating the range of the low-pass filter.

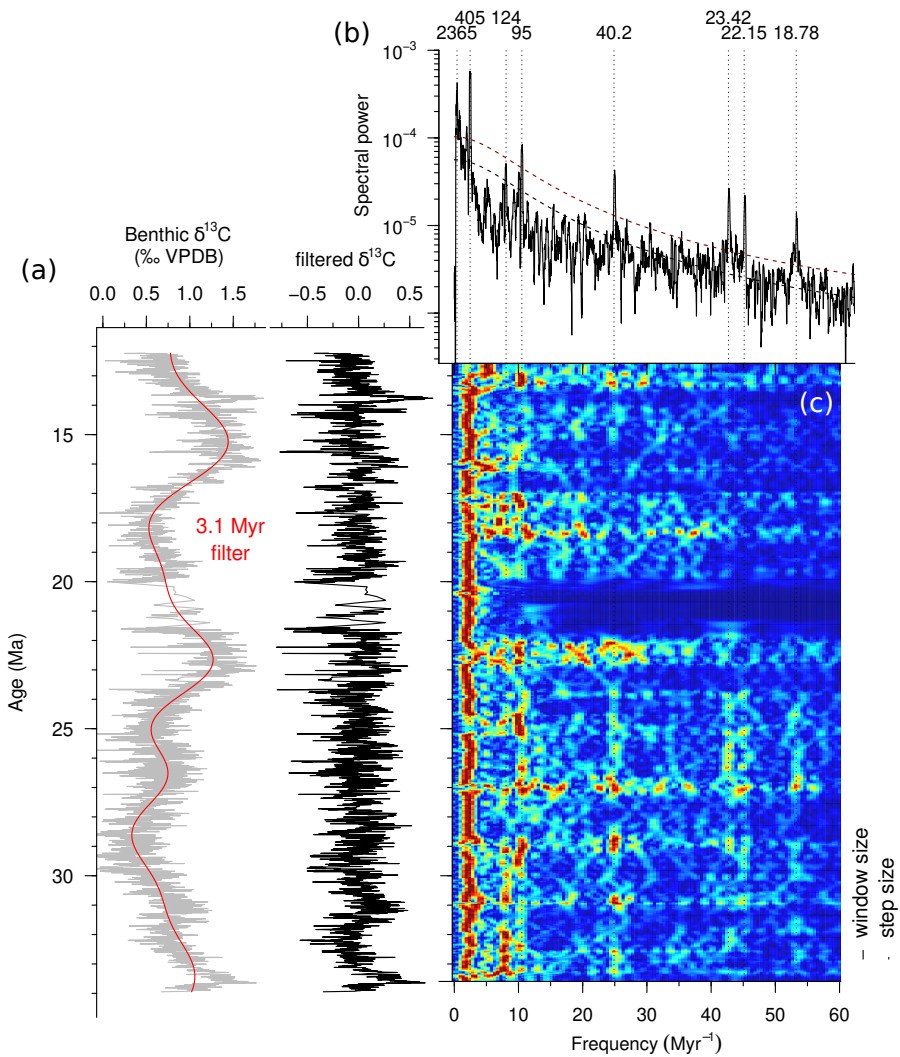

**Figure 6.** Evolutive multitaper harmonics analysis (c) of the data composite $\delta^{13}$C (grey) after detrending (black) the $<1$ Myr signal (red line) in the time (a), frequency (b) domain. Periods of interest are labelled (b) at the same axis as (c). A window size of $0.7$ Myr was used, with a step size of $10$ kyr (lines on the right side in (c)). Methods adapted from Pälike et al. (2006).