# Peer review of "The 405 kyr and 2.4 Myr eccentricity components in Cenozoic carbon isotope records"

_Climate of the Past, 2018_

## Referee Comment (RC1) · Anonymous Referee #1 · 26 May 2018

General comments:

This paper explores the relationship between orbital forcing and global carbon cycles across different timescales. For this purpose, the authors simulated the response of several key components of the carbon cycles to the orbital forcing using the LOSCAR model and compared those to the composite benthic foraminiferal d13C and d18O records covering the Eocene to Miocene. They highlighted the potential dominance of 2.4 Myr eccentricity cycle in the model results and argue that this cycle maybe overshadowed by the long-term changes in the composite records.

Overall, the structure of this paper is well shaped and data are clearly presented. However, I have two main concerns that need to be clarified by the authors: As suggested by the authors in the introduction of this paper, the co-occurrence of 405 and 100 kyr

eccentricity maxima and relatively negative d13C and d18O values is still elusive, and different mechanisms, such as organic carbon burial linked to weathering controlled clay deposition, variations of global exogenic carbon pool, submarine methane hydrate dynamics, etc, have been suggested to address this. However, without any further discussion of these different mechanisms in the latter part of this paper, it seems to me that the authors already think the organic carbon burial plays a very important role in this co-occurrence before testing it, which doesn't make sense to me. Moreover, the authors argue that the continental shelves are very important for the organic carbon storage over astronomical time scales, which leads them to linearly force the Corg burial in marine sediment with astronomical forcing in their modelling. This doesn't make sense to me as both aspects, on the one hand, how astronomical cycles force the changes in the continental shelves, what the role of weathering has played, and why it is linear relationship between them, on the other hand, how burial of organic matter is regulated in the continental shelves, are still open questions. The authors found strong 2.4 Myr cyclicity in their model output but no signal in the composite records based on the MTM spectral analysis, which they argue the 2.4 Myr cyclicity may act purely as amplitude modulator, not as a true cycle. To me it is very important to address why 2.4 Myr cyclicity is dominant in the model output but act as an AM in the composite records as this is the key to understand the questions raised by the authors. Although the author expect to evaluate this with high-resolution proxy records in the future, I feel this part is under-discussed since this is certainly the highlight of this manuscript.

I also have a few technical comments:

1) Labels for Figure 2 are missing. 2) In model-data comparison, the authors said 'the lag of d13C to the 405 kyr eccentricity cycle is ∼190 kyer, which is much higher than the 40 to 60 kyr during the Paleocene', it would be better for the authors to address the phase relationship between different records using cross spectral analysis, such as 'Application of the cross wavelet transform and wavelet coherence to geophysical time series' by Grinsted et al., 2004. 3) For Figure 4, I recommend using the wavelet analysis to assess whether this lead-lag relationship is constant through time, see Grinsted et al., 2004 for details.

---

## Referee Comment (RC2) · Anonymous Referee #2 · 5 Jun 2018

This manuscript presents an interesting modelling study of the eccentricity components in the Cenozoïc carbon isotope records. Overall, the paper is well written and the obtained results are worth publishing. I nevertheless have some major comments on the model-data comparison and the overall discussion, which both appear quite incomplete and insufficient.

Major comments

1 - The model used here computes many variables, but only a few of them are discussed. In particular, since the ocean temperatures (in the Pacific) are computed, I would expect to find a comparison between them and the 18O records presented in the introduction (Figure 1). Such a comparison is not shown and not even discussed. From the legend of Figure 2 (Ân deep Pacific temperature . . . are omitted from this plot

because their pattern mirror that of Ctot), I understand that temperature is dominated by the carbon forcing (pCO2) and exhibits a strong 2.4 Myr oscillation. What about its amplitude ? Is it compatible with data ? I suspect this is not the case. The pCO2 changes are large (about a pCO2 doubling at 2.4 Myr frequency) therefore modelled temperature changes should be in the 3 to 4°C range for rather standard values for climate sensitivity, and the corresponding 18O amplitude should be in the 0.5 to 1‰ range. This does not seem to be compatible with data as shown on Figure 1. In any case, whatever the results, I do not understand the modelling strategy: why using a rather sophisticated model that computes many outputs, but only discussing (cherry picking ?) some of them, and not others ?

2 – Similarly, the CCD output of the model is also not compared to the real world. This is a bit less disturbing, since the authors are not showing any CCD reconstructions over this time span. But such data exist, though probably with low resolution. For instance, according to Paelike et al (Nature, 2012), the Pacific CCD is deep and stable during this time (Oligocene-Miocene) up to about 18.5 Ma. Again, this does not seem consistent with the model results. Interestingly, they also observe rather large changes in the CCD during the Eocene (CAE events), that may, or may not, correspond to the simulated changes in this paper ? In any case, some thorough discussion of the CCD outputs with respect to observations seems to me absolutely necessary. Again, what is the point of computing these variables, if not for performing some comparison with data ?

3 – According to the final sentence, high resolution pCO2 proxies would be necessary to check if a strong 2.4 Myr signal is present or not over the Cenozoïc, as suggested by the model. As mentioned in my first comment, such a strong eccentricity signal in greenhouse gases should probably have already been detected in the (climate) 18O data. But more importantly, such pCO2 proxy data are in fact readily available for the Plio-Pleistocene (eg. Bartoli et al., 2011 ; Seki et al, 2010 ; …) and no such large (2x) pCO2 changes are seen, while large climatic changes are obvious. A simple model

also based on organic carbon burial on continental margins was proposed recently (Paillard, CP, 2017) to account both for the 13C and the pCO2 data over this period.

4 – Spectral analysis represents a rather large part of the paper (Figs 3 to 6). Still, I do not quite understand how this helps for the discussion or for the conclusions, beyond the (quite expected) fact that long-term processes are acting as low-pass filters. Either I missed some important point, or probably there is a far too large weigh on spectral analysis in this paper.

5 – The same comment applies for the use of red noise in the experiments. Obviously, there is here some (rather implicit) attempt to Âń fit Âż the spectrum of the data with a deterministic model plus a red noise. But what is the point ? I understand that red noise fitting is useful for spectral line detection. But this is not the topic of the paper. What do the noisy experiments tell us about the dynamics of organic carbon on continental margins ? Does a nice spectral fit help the authors to make their point ? I am personally not convinced.

Other comments

6 – Page 4 line 18 ; Page 10 line 3 and line 10 Âń shifting of spectral power Âż. I certainly do not recommend using this word Âń shifting Âż in the current context. In signal processing, a spectral shift is a change of frequency between input and output. This is not the case here, since output frequencies are exactly the same as input ones. The authors are referring possibly to the fact that high frequencies are damped. This is simply called a low-pass filter, not a frequency shift. Or possibly that the amplitude modulation of the forcing can be extracted thanks to some non-linearity of the model (or "clipping" of the forcing). But again, this is not a "shift" in frequency, but a de-modulation (ie. the most simple tone-combination).

7 – Page 7, line 7 : ETP Âń could be considered more objective Âż. I do not understand why. From a mathematical viewpoint, using an insolation forcing is a parametric choice (eg. choosing latitude and season). ETP is also parametric (relative weigh of tilt and

precession). Furthermore, in ETP, the phasing of precession is arbitrarily fixed to 2 values only (a plus or a minus sign), while choosing a specific season offers more freedom. Insolation is more physically based, ETP is not. Obviously, there is no specific reason to choose 65°N in summer, since there are no ice sheet present at this location. In the context of the model presented here, a much more simple and objective choice would be to use only one parameter : eccentricity, or tilt, or precession only. Then the discussion on mechanisms would be easier. Since the focus is on the 405 kyr and 2.4 Myr eccentricity frequencies in the carbon system, I am not sure that using tilt in the forcing is relevant (except may be to discuss the 1.2 Myr modulation vs the 2.4 Myr eccentricity one, but this is probably not quite the topic).

8 – Legend of Figure 2 : atmospheric $pCO_2$ is shown on the figure, though the legend says just the opposite.

9 – Page 10 line 17 : Âń 180° . . . out-of-phase Âż. No, since Âń out-of-phase Âż means that there is no phase relationship. Here, it is in Âń anti-phase Âż, something very similar to Âń in-phase Âż.

10 – Page 11 line 9 : Âń $\delta$13C minima are phased with ETP maxima Âż. No. The authors probably mean that $\delta$13C minima are compared with ETP maxima, or equivalently that the relationship between $\delta$13C and ETP is measured by the phasing between -$\delta$13C and ETP.

11 – Page 11 line 26 : Âń 0.5 Myr Âż please add exponent -1.

12 – Many figure legends in the supplement are incomplete or inconsistent with the figure. which makes it difficult to understand... Figure 1: 3 colored curves but only 2 description. Figure 2: A/The red and purple curves are co2 and pacific temperature, but which one is which ? According to B, purple might be CO2, but what is "tcb" ? Figure 3: What is the green curve ? Figure 6: The legend mentions 18O records, but the figures displays apparently only 13C ones

References

Pälike H et al. A Cenozoic record of the equatorial Pacific carbonate compensation depth. Nature (2012) vol. 488 (7413) pp. 609-614. Bartoli G et al. Atmospheric CO2 decline during the Pliocene intensification of Northern Hemisphere glaciations. Paleoceanography (2011) vol. 26 (4). Seki et al. Alkenone and boron-based Pliocene pCO2 records. Earth Planet. Sci. Lett. (2010) vol. 292 (1-2) pp. 201-211. Paillard D. The Plio-Pleistocene climatic evolution as a consequence of orbital forcing on the carbon cycle. Cli

---

## Author Comment (AC2) · 19 Jul 2018

We thank reviewer 2 for his/her constructive and elaborate feedback on the manuscript.

Reply to major comments

1) Our study explores the behaviour of the well-documented LOSCAR model (e.g. Zeebe 2012 GMD) under orbital forcing. This carbon cycle model is rather simple and generates multiple output variables. In our initial paper we showed all the tracers relevant for this study. In the revised version we will add all model output in the digital supplement for completeness: it will allow the reader to evaluate alternatives. Detailing the behaviour of all tracers in the main text would not only dilute the main message, but it would also largely repeat what has been published before.

[Figure]

The model calculates temperature (TCB in the supplementary plot, we shall update the legend) from atmospheric $CO_2$ from an input temperature sensitivity parameter of 5 °C, resulting in changes of about 3 °C (as estimated by the reviewer). Thus, if we simply translate these into $\delta^{18}O$, this would result in 2.4 Myr cyclicity with a too-high amplitude (~0.75 ‰ when compared to our data-composite.

However, it is important to note that the LOSCAR model is a carbon cycle model and not a climate model. Thus, its simple climate sensitivity equation relates $CO_2$ and temperature to each other directly, whereas in reality temperature is a function of a number of variables, including but not limited to $pCO_2$. For instance, insolation directly affects the climate system components (e.g. atmosphere, etc.), which in turn affect temperature. These insolation-driven changes to the climate system are not part of the temperature-response to $CO_2$, however. This behaviour cannot be captured by the carbon cycle model. The above justifies our initial focus on $\delta^{13}C$ in the data–model comparison, rather than $\delta^{18}O$. Therefore it is not surprising that temperature, $\delta^{18}O$, and $pCO_2$ records show different power spectra, whereas our carbon cycle model output does not. Thus the lack of a strong 2.4 Myr cycle in $\delta^{18}O$ (and temperature) records does not necessarily suggest that the 2.4 Myr cycle can not be dominant in $pCO_2$.

Furthermore, the $pCO_2$ records that the referee refers to (e.g. Bartoli et al. 2012, Seki et al. 2010) have neither the length nor the resolution to pin down a possible 2.4 Myr cycle. New, long $pCO_2$ proxy records are required to establish a possible 2.4 Myr cyclicity.

However, because qualitative comparison of model output to $\delta^{18}O$ records would be useful to the reader, we shall add the analysis to the model–data comparison in the results and discussion, and further elaborate on the fact that indeed, the model shows strong 2.4 Myr cyclicity, whereas the record shows very weak cyclicity (possibly over-shadowed) as well as amplitude modulation (AM). The above possible explanation for this discrepancy will be introduced in the discussion.

2) Comparing the CCD model output changes to those reconstructed by Pälike et al. (2012) would be a great addition to the manuscript. In terms of absolute values this exercise would not be very informative, however, since it is dependent on multiple input parameters of the model. The main aim of this study is not necessarily to get the best agreement between model predictions and data based reconstructions, but rather to study the possible underlying mechanisms.

As to whether strong 2.4 Myr cyclic fluctuations in the CCD are realistic, this is rather hard to estimate from the data. Based on this suggestion by the reviewer we have revisited several CaCO$_3$ datasets, including those published by Pälike et al. (2012). We have attempted to perform spectral analysis on the CCD reconstructions from Pälike et al. (2012), but the highest resolution record (Site U1334) has a sampling resolution of ~85 kyr, meaning that it would barely be able to resolve 425 kyr cyclicity without the risk of aliasing. We find no significant longer periods in their data.

3) We shall add the $\delta^{18}$O of the composite record to the analysis in the supplement, to facilitate model–data comparison. Additionally, we shall comment on the likeliness of the $\delta^{18}$O record being reflective of CO$_2$ dynamics in the discussion. We thank the referee for bringing the very relevant Paillard (2017) paper to our attention. It is a beautiful extra motivation for our study and we will acknowledge and address it as such revised manuscript.

4) Spectral analysis indeed forms the backbone of this paper, as it is the primary tool by which we assess the presence and impact of the 2.4 Myr and 405 kyr cycles. We deliberately included detailed spectral analysis based on multiple techniques, to inform the reader about, a.o. the spectra of different model (and data) variables, amplitude modulation thereof and cross-correlation between variables.

5) In our simulations, we use white noise as input, which results in red-noise in model output. The goal of adding relatively a lot of white noise to the astronomical forcing (50/50 noise/signal), is to assess how our simulated signal could be perturbed by unknown/stochastic processes. We agree that creating a nice spectral fit to the data is of a secondary/non-pertinent nature and not the main focus of our study. We will modify the text to articulate this.

Other comments

6) We thank the referee for clarification. We shall rephrase 'shifting of spectral power' to low-pass filtering.

7) We agree, we shall adjust our wording in these lines and include separate model runs for eccentricity and precession to the supplement and include discussion of these in the main text discussion. For completeness, runs with 65°N and 30°N insolation curves as forcing will also be added to the supplement (see attached figures for initial drafts of these figures).

8) We shall revise the legend.

9) We shall revise out-of-phase to in anti-phase.

10) We shall revise the wording to more clearly reflect our approach.

11) Exponent $-1$ will be added.

12) Figures in the supplement.

Figure 1: These are actually not three colours, but two lines with low opacity. We shall choose different colours to improve readability of the figure.

Figure 2: red = temperature, purple = $CO_2$. We shall change the legend in B.

Figure 3: The purple curve is the data composite $\delta^{13}C$, while the green curve is $\delta^{13}C$ model output for the specified run. The legend will be updated.

Figure 6: We shall re-add the $\delta^{18}O$ records to the figure.
* * *
[Figure]

**Fig. 1.** A time-series with eccentricity as forcing. Thick lines through the tracers represent 405 kyr and 2.4 Myr filters.

**Fig. 2.** A time-series with precession as forcing.

[Figure]

**Fig. 3.** A time-series with 30°N summer insolation as forcing.

**Fig. 4.** A time-series with 65°N summer insolation as forcing. Notice that the long-term trend is the result of the long-term trend in obliquity (see Laskar et al. 2004, Science, fig. 14)

[Figure]

---

## Author Response (AR2)

**1  Editor comments**

Dear authors,

I first would like to apologize for the delay in the review process. The two reviewers acknowledge the interest of the study, but they both raise questions preventing the acceptance of your manuscript without major revisions. Among the key questions raised, I think that the hypothesis of linearity dependence of organic carbon burial on astronomical forcing is far from being obvious, and impact strongly your results. This should be extensively discussed in the revised version, with adequate references. The other reviewer raised the question of existence of the $p\,CO_2$ high resolution data for the Plio-Pleistocene period that should be included in your study. These two points, together with all the other questions, should be answered or discussed in the revised manuscript, that I encourage you to submit. Once the revised version received, it will go for a second round of reviews.

Best regards.

Dear editor,
We think we have addressed the concerns about our choice for linear forcing and the available $p\,CO_2$ and CCD records in the revised manuscript. Below, you will find our responses to all the points by the reviewers, followed by a marked-up document with changes relative to the first version (removed words in red, added words in blue).
We look forward to receiving more feedback on the document.
Kind regards,
on behalf of all co-authors,
Ilja Kocken

**2  Referee 1**

General comments:

This paper explores the relationship between orbital forcing and global carbon cycles across different timescales. For this purpose, the authors simulated the response of several key components of the carbon cycles to the orbital forcing using the LOSCAR model and compared those to the composite benthic foraminiferal $\delta^{13}C$ and $\delta^{18}O$ records covering the Eocene to Miocene. They highlighted the potential dominance of 2.4 Myr eccentricity cycle in the model results and argue that this cycle maybe overshadowed by the long-term changes in the composite records.

We thank the reviewer for his/her feedback on the manuscript.

Overall, the structure of this paper is well shaped and data are clearly presented. However, I have two main concerns that need to be clarified by the authors: As suggested by the authors in the introduction of this paper, the co-occurrence of 405 and 100 kyr eccentricity maxima and relatively negative $\delta^{13}$C and $\delta^{18}$O values is still elusive, and different mechanisms, such as organic carbon burial linked to weathering controlled clay deposition, variations of global exogenic carbon pool, submarine methane hydrate dynamics, etc, have been suggested to address this. However, without any further discussion of these different mechanisms in the latter part of this paper, it seems to me that the authors already think the organic carbon burial plays a very important role in this co-occurrence before testing it, which doesn't make sense to me. Moreover, the authors argue that the continental shelves are very important for the organic carbon storage over astronomical time scales, which leads them to linearly force the Corg burial in marine sediment with astronomical forcing in their modelling. This doesn't make sense to me as both aspects, on the one hand, how astronomical cycles force the changes in the continental shelves, what the role of weathering has played, and why it is linear relationship between them, on the other hand, how burial of organic matter is regulated in the continental shelves, are still open questions. The authors found strong 2.4 Myr cyclicity in their model output but no signal in the composite records based on the MTM spectral analysis, which they argue the 2.4 Myr cyclicity may act purely as amplitude modulator, not as a true cycle. To me it is very important to address why 2.4 Myr cyclicity is dominant in the model output but act as an AM in the composite records as this is the key to understand the questions raised by the authors. Although the author expect to evaluate this with high-resolution proxy records in the future, I feel this part is under-discussed since this is certainly the highlight of this manuscript.

We introduce many possible mechanisms for the co-occurrence of the 405 and 100 kyr eccentricity maxima in climate records, since they are still poorly understood. We then dive into one of the possible mechanisms, using a relatively simple forcing scheme – linear forcing of organic carbon (OC) burial in the continental shelves. Since the mechanisms are largely unknown, but we do know that they must link astronomical forcing to changes in $\delta^{13}$C and $\delta^{18}$O, we decide to force OC burial directly, and linearly, as a first approximation. OC burial is the product of sediment accumulation and organic carbon contents (corrected for porosity) and is thus linearly related to sediment accumulation, which in turn is approximately linearly related to sediment delivery to the ocean in the absence of major changes in sea level. Other factors governing OC burial (oxygen etc.) are secondary to sediment deposition. Moreover, we believe that our linear approach is preferred as a starting point, and find that it is quite capable of explaining some of the patterns we observe in climate records. We

have have added a paragraph to explain this better to the introduction.

Secondly, we have elaborated on the significance between the discrepancy between model output (2.4 Myr as a cycle) and data composite (2.4 Myr a amplitude modulation (AM)) in the model–data comparison section. Mainly, we do not expect to find AM in the model output, since there is no mechanism in the model that would introduce such forcing and there is no reason to expect it to show as an emergent property. This means that in the real world, a certain process or processes causes the 2.4 Myr cycle to be present as an AM of the shorter eccentricity cycles, that we apparently do not incorporate in this simple carbon cycle model. At this point, with the available information, it is difficult to speculate regarding the exact causes of this result although we very much agree with the reviewer that it is an important result.

I also have a few technical comments:

1. Labels for Figure 2 are missing. 2) In model-data comparison, the authors said 'the lag of $\delta^{13}$C to the 405 kyr eccentricity cycle is ~190 kyr, which is much higher than the 40 to 60 kyr during the Paleocene', it would be better for the authors to address the phase relationship between different records using cross spectral analysis, such as 'Application of the cross wavelet transform and wavelet coherence to geophysical time series' by Grinsted et al., 2004. 3) For Figure 4, I recommend using the wavelet analysis to assess whether this lead-lag relationship is constant through time, see Grinsted et al., 2004 for details.

1. We have added the a–d labels.

2. Indeed, we do use cross-spectral analysis using the Blackman-Tukey (BT) method from Analyseries to calculate this 190 kyr lag in the model. Using the suggested wavelet-based analysis would allow us to estimate changes in this relationship through time. However, we have no reason to suspect that the model output will show different phases in different time-slices, since it is forced with a simple cyclic input. If we look at the model output time-series, there is also no reason to suspect temporal changes in the phasing. The estimates of 40 to 60 kyr for the lag in the data are from the literature.

   We are open to applying cross-wavelet analysis, but we do not see the added value. We tried to access the Cross-Wavelet matlab script the reviewer pointed us to (from `http://www.pol.ac.uk/home/research/waveletcoherence/`) but this script was not accessible.

3. We could attempt to perform Cross-Wavelet analysis using the R-package WaveletComp (`http://www.hs-stat.com/projects/WaveletComp/WaveletComp_guided_tour.pdf`), but as mentioned before, we do not see the added value and are open to further editorial advice on this particular issue.

**3 Referee 2**

> This manuscript presents an interesting modelling study of the eccentricity components in the Cenozoic carbon isotope records. Overall, the paper is well written and the obtained results are worth publishing. I nevertheless have some major comments on the model–data comparison and the overall discussion, which both appear quite incomplete and insufficient.

We thank reviewer 2 for his/her constructive and elaborate feedback on the manuscript.

> Major comments
>
> 1 - The model used here computes many variables, but only a few of them are discussed. In particular, since the ocean temperatures (in the Pacific) are computed, I would expect to find a comparison between them and the $^{18}$O records presented in the introduction (Figure 1). Such a comparison is not shown and not even discussed.

Our study explores the behaviour of the well-documented LOSCAR model (e.g. Zeebe 2012 GMD) under orbital forcing. The climate parametrizations in this carbon cycle model are rather simple and generate multiple output variables. In our initial submission we showed all the tracers relevant for this study. In the revised version we have added all model output in the digital supplement for completeness: it will allow the reader to evaluate alternatives. Detailing the behaviour of all tracers in the main text would not only dilute the main message, but it would also largely repeat what has been published before.

> From the legend of Figure 2 (deep Pacific temperature ... are omitted from this plot because their pattern mirror that of $\{C_{tot}\}$), I understand that temperature is dominated by the carbon forcing ($p\,CO_2$) and exhibits a strong 2.4 Myr oscillation. What about its amplitude? Is it compatible with data? I suspect this is not the case. The $p\,CO_2$ changes are large (about a $p\,CO_2$ doubling at 2.4 Myr frequency) therefore modelled temperature changes should be in the 3 to 4 °C} range for rather standard values for climate sensitivity, and the corresponding $^{18}$O amplitude should be in the 0.5 to 1 ‰ range. This does not seem to be compatible with data as shown on Figure 1. In any case, whatever the results, I do not understand the modelling strategy: why using a rather sophisticated model that computes many outputs, but only discussing (cherry picking?) some of them, and not others?

The model calculates temperature (TCB in the supplementary plot, we have updated the legend to "temperature") from atmospheric $CO_2$ from an input temperature sensitivity parameter of 5 °C, resulting in changes of about 3 °C as estimated by the reviewer). Thus, if we simply translate these into $\delta^{18}$O, this

would result in 2.4 Myr cyclicity with a too-high amplitude ($\sim$0.75 ‰) when compared to our data-composite.

However, it is important to note that the LOSCAR model is a carbon cycle model and not a climate model. Thus, its simple climate sensitivity equation relates $CO_2$ and temperature to each other directly, whereas in reality temperature is a function of a number of variables, including but not limited to $p\,CO_2$. For instance, insolation directly affects the climate system components (e.g. atmosphere, etc.), which in turn affect temperature. These insolation-driven changes to the climate system are not part of the temperature-response to $CO_2$, however. This behaviour cannot be captured by the carbon cycle model. The above justifies our initial focus on $\delta^{13}C$ in the data–model comparison, rather than $\delta^{18}O$. Therefore it is not surprising that temperature, $\delta^{18}O$, and $p\,CO_2$ records show different power spectra, whereas our carbon cycle model output does not. Thus the lack of a strong 2.4 Myr cycle in $\delta^{18}O$ (and temperature) records does not necessarily suggest that the 2.4 Myr cycle can not be dominant in $p\,CO_2$.

We have added the $\delta^{18}O$ of the composite record to the analysis in the supplement, to facilitate model–data comparison (supplement fig. 6), and added a paragraph to elaborate on this to the model–data comparison section of the results and discussion.

> 2 – Similarly, the CCD output of the model is also not compared to the real world. This is a bit less disturbing, since the authors are not showing any CCD reconstructions over this time span. But such data exist, though probably with low resolution. For instance, according to Päelike et al (Nature, 2012), the Pacific CCD is deep and stable during this time (Oligocene-Miocene) up to about 18.5 Ma. Again, this does not seem consistent with the model results. Interestingly, they also observe rather large changes in the CCD during the Eocene (CAE events), that may, or may not, correspond to the simulated changes in this paper? In any case, some thorough discussion of the CCD outputs with respect to observations seems to me absolutely necessary. Again, what is the point of computing these variables, if not for performing some comparison with data?

Comparing the CCD model output changes to those reconstructed by Päelike et al. (2012) would be a nice addition to the manuscript. In terms of absolute values this exercise would not be very informative, however, since it is dependent on multiple input parameters of the model. The main aim of this study is not necessarily to get the best agreement between model predictions and data based reconstructions, but rather to study the possible underlying mechanisms.

We have included a section on this in the model–data comparison section. While we were able to perform some analysis with moderate success, as described in the revised manuscript, problems arise due to problems with the age-models for the Eocene. Resolving these problems for all individual records is beyond the scope of this manuscript.

3 – According to the final sentence, high resolution $p\,CO_2$ proxies would be necessary to check if a strong 2.4 Myr signal is present or not over the Cenozoic, as suggested by the model. As mentioned in my first comment, such a strong eccentricity signal in greenhouse gases should probably have already been detected in the (climate) $^{18}$O data. But more importantly, such $p\,CO_2$ proxy data are in fact readily available for the Plio-Pleistocene (eg. Bartoli et al., 2011; Seki et al, 2010; ...) and no such large $(2\times)$ $p\,CO_2$ changes are seen, while large climatic changes are obvious. A simple model also based on organic carbon burial on continental margins was proposed recently (Paillard, CP, 2017) to account both for the $^{13}$C and the $p\,CO_2$ data over this period.

The $p\,CO_2$ records that the referee refers to (e.g. Bartoli et al. 2012, Seki et al. 2010) have neither the length nor the resolution to pin down a possible 2.4 Myr cycle, but we have added a short paragraph on them to the discussion. New, long $p\,CO_2$ proxy records are required to establish a possible 2.4 Myr cyclicity.

We have commented on the likeliness of the $\delta^{18}$O record being reflective of $CO_2$ dynamics in the discussion. We thank the referee for bringing the very relevant Paillard (2017) paper to our attention. It is a relevant reference for our study and we have acknowledged and addressed it as such in the revised manuscript.

4 – Spectral analysis represents a rather large part of the paper (Figs 3 to 6). Still, I do not quite understand how this helps for the discussion or for the conclusions, beyond the (quite expected) fact that long-term processes are acting as low-pass filters. Either I missed some important point, or probably there is a far too large weigh on spectral analysis in this paper.

Spectral analysis indeed forms the backbone of this paper, as it is the primary tool by which we assess the presence and impact of the 2.4 Myr and 405 kyr cycles. We deliberately included detailed spectral analysis based on multiple techniques, to inform the reader about, *inter alia* the spectra of different model (and data) variables, amplitude modulation thereof and cross-correlation between variables.

5 – The same comment applies for the use of red noise in the experiments. Obviously, there is here some (rather implicit) attempt to fit the spectrum of the data with a deterministic model plus a red noise. But what is the point? I understand that red noise fitting is useful for spectral line detection. But this is not the topic of the paper. What do the noisy experiments tell us about the dynamics of organic carbon on continental margins? Does a nice spectral fit help the authors to make their point? I am personally not convinced.

In our simulations we use white noise as input which results in red-noise in model output. The goal of adding relatively strong white noise to the astronomical forcing (50/50 noise/signal), is to assess how our simulated signal could be perturbed by unknown/stochastic processes. We agree that creating a nice spectral fit to the data is of a secondary/non-pertinent nature and not the main focus of our study. We have modified the material and methods section to articulate this.

> Other comments
>
> 6 – Page 4 line 18; Page 10 line 3 and line 10 shifting of spectral power. I certainly do not recommend using this word shifting in the current context. In signal processing, a spectral shift is a change of frequency between input and output. This is not the case here, since output frequencies are exactly the same as input ones. The authors are referring possibly to the fact that high frequencies are damped. This is simply called a low-pass filter, not a frequency shift. Or possibly that the amplitude modulation of the forcing can be extracted thanks to some non-linearity of the model (or "clipping" of the forcing). But again, this is not a "shift" in frequency, but a de-modulation (ie. the most simple tone-combination).

We thank the referee for clarification. We have rephrased 'shifting of spectral power' to low-pass filtering throughout the manuscript.

> 7 – Page 7, line 7 : ETP could be considered more objective . I do not understand why. From a mathematical viewpoint, using an insolation forcing is a parametric choice (eg. choosing latitude and season). ETP is also parametric (relative weigh of tilt and precession). Furthermore, in ETP, the phasing of precession is arbitrarily fixed to 2 values only (a plus or a minus sign), while choosing a specific season offers more freedom. Insolation is more physically based, ETP is not. Obviously, there is no specific reason to choose 65°N in summer, since there are no ice sheet present at this location. In the context of the model presented here, a much more simple and objective choice would be to use only one parameter: eccentricity, or tilt, or precession only. Then the discussion on mechanisms would be easier. Since the focus is on the 405 kyr and 2.4 Myr eccentricity frequencies in the carbon system, I am not sure that using tilt in the forcing is relevant (except may be to discuss the 1.2 Myr modulation vs the 2.4 Myr eccentricity one, but this is probably not quite the topic).

We agree, we have adjusted our wording in these lines and included separate model runs for eccentricity and precession to the supplement (suppl. fig. 9) and have included discussion of these in the main text discussion. For completeness, runs with 65°N and 30°N insolation curves as forcing were also added to the supplement (suppl. fig. 9).

8 – Legend of Figure 2: atmospheric $p\,CO_2$ is shown on the figure, though the legend says just the opposite.

We have revised the legend.

9 – Page 10 line 17: 180° ... out-of-phase. No, since out-of-phase means that there is no phase relationship. Here, it is in anti-phase, something very similar to in-phase.

We have revised out-of-phase to in anti-phase.

10 – Page 11 line 9: $\delta^{13}C$ minima are phased with ETP maxima. No. The authors probably mean that $\delta^{13}C$ minima are compared with ETP maxima, or equivalently that the relationship between $\delta^{13}C$ and ETP is measured by the phasing between $\delta^{13}C$ and ETP.

We have revised the wording to more clearly reflect our approach.

11 – Page 11 line 26 : 0.5 Myr please add exponent -1.

Exponent -1 was added.

12 – Many figure legends in the supplement are incomplete or inconsistent with the figure. which makes it difficult to understand... Figure 1: 3 colored curves but only 2 description. Figure 2: A/The red and purple curves are $CO_2$ and pacific temperature, but which one is which? According to B, purple might be $CO_2$, but what is "tcb"? Figure 3: What is the green curve? Figure 6: The legend mentions $^{18}O$ records, but the figures displays apparently only 13 C one.

Figure 1: These are actually not three colours, but two lines with low opacity. We have chosen different colours to improve readability of the figure.
Figure 2: red = temperature, purple = $CO_2$. We have changed the legend in (b).
Figure 3: The purple curve is the data composite $\delta^{13}C$, while the green curve is $\delta^{13}C$ model output for the specified run. The colours have been changed and the legend has been updated.
Figure 6: We have re-added the $\delta^{18}O$ records to the figure.

References:

Pälike H et al. A Cenozoic record of the equatorial Pacific carbonate compensation depth. Nature (2012) vol. 488 (7413) pp. 609-614. Bartoli G et al. Atmospheric $CO_2$ decline during the Pliocene intensification of Northern Hemisphere glaciations. Paleoceanography (2011) vol. 26 (4). Seki et al. Alkenone and boron-based Pliocene $p\,CO_2$ records. Earth Planet. Sci. Lett. (2010) vol. 292 (1-2) pp. 201-211. 
[revised manuscript text omitted]

**2 Calcite compensation depth spectral analysis**

[Figure]

**Figure 1.** Carbonate accumulation rate (CAR) in the time (left side) and frequency domains (right side) for all sites in Pälike et al. (2012). Loess filtered (purple) and unfiltered (black) MTM-spectra are shown (right side) for the selected time-slices (darker dots in left side). Time-slices were selected based on continuous similar-resolution intervals where there are no known time-scale issues. Corresponding significant (harmonic f-test) peaks are labelled with associated periods.

[Figure]

**Figure 2.** CCD as reconstructed by Pälike et al. (2012) (a), filtered for 405 kyr and 2.4 Myr (b) with a loess fit (a, blue line) subtracted from the record (c), which result in power spectra for the original (gray) and detrended (black) Pacific CCD composite (c). Significant peaks are labelled with the corresponding periods.

**Table 1.** Periods of interest in the input forcing were calculated from dominant secular frequencies (g, from Laskar et al. (2004)). Non-eccentricity typical periods were found visually with by analysing in detail the MTM-spectra of component. The index $i$ in $g_i$ refers to the planet in the solar system (counting from the sun outward).

| Component | Terms | Period (yr) | Plot period (kyr) |
|---|---|---|---|
| Eccentricity | $g_4 - g_3$ | 2365408.22085889 | 2365 |
| Eccentricity | $g_1 - g_5$ | 972584.643914338 | 973 |
| Eccentricity | $g_2 - g_1$ | 696021.261160021 | 696 |
| Eccentricity | $g_2 - g_5$ | 405691.71449262 | 405 |
| Eccentricity | $g_2 - g_4$ | 123854.477912872 | 124 |
| Eccentricity | $g_4 - g_5$ | 94886.4059134611 | 95 |
| Obliquity | | | 52.4 |
| Obliquity | | | 40.2 |
| Obliquity | | | 29.5 |
| Obliquity | | | 28.5 |
| Precession | | | 23.42 |
| Precession | | | 22.15 |
| Precession | | | 18.78 |
| Precession | | | 16.34 |

**3 Effects of shifting records for consistency in overlap**

Attempts were made to improve inter-record compatibility by shifting record $\delta^{13}C$ and $\delta^{18}O$ values from the various studies towards heavier or lighter values, such that mean isotope values in overlapping regions would be identical, but were eventually abandoned because of their minor influence on spectral outcome ( Supp. Fig. 3).

[Figure]

**Figure 3.** A linear shift in the record by Pälike et al. (2006) (down by $0.30‰$ VPDB in red, original in blue) increases consistency between records in the area of overlap (where data from Pälike et al. (2006) were omitted due to lower resolution) but hardly effects spectral outcome and is therefore not used in this study.

**Table 2.** Effects of increasing various parameters on model output.

| Increased parameter | Causes this effect |
| --- | --- |
| initial pCO$_2$ | Base level of $\delta^{13}$C increases |
| orbstrength | Amplitude of cycle variation in tracers increases |
| cliplevel | Increased shift in spectral power to lower frequencies |
| E, T and P in ETP | Shift spectral power to associated frequencies |
| noiselevel | Amplitude of noise increases, base of AR-1 fit in MTM-spectra is raised |
| noisetiming | More noise is generated in lower frequency range |

**4  Model sensitivity**

The introduction of white noise on C$_{org}$ burial results in red noise in the spectral output of the model tracers. This basically means that the low-frequency spectral power amplification that we see in the orbitally-forced runs also occurs for the noise input. The addition of noise thus adds spectral power to the low-frequency range, and raises the background levels of MTM-spectra in the orbitally-forced model output tracers.

The model responds as expected to changes in initial conditions and newly added parameters. Table 2 shows an overview of what happens to the model output when a parameter is increased.

[Figure]

**Figure 4.** Model output when  C_org burial is only affected by noise that is generated every time a time step larger than 1 kyr has passed in the time (A) and frequency (B) domain.

[Figure]

**Figure 5.**  Comparison of the detrended data composite of $\delta^{13}$C (a) and $\delta^{18}$O (b) to model output $\delta^{13}$C (c) and $p$CO₂ (d) forced with ETP (1:0.5:−0.4) with an  `orbstrength` of  0.5 and `noiselevel` of 0.2. 2.4 Myr and 405 kyr bandpass filters are shown in purple (shifted up by arbitrary amount).

**5 Apparent lead in $2.4\,\mathrm{Myr}$ of the $-\delta^{13}C_{DP}$ signal**

[Figure]

**Figure 6.** Transient linearly increasing orbital forcing (orange line) and the response of $\delta^{13}C$ (green line) and pCO$_2$ (purple line). Horizontal dotted lines are visual aids to track the initial values of $\delta^{13}C$ and pCO$_2$. The double arrow marked "$\delta^{13}C$ overshoot" is drawn between the initial value of $\delta^{13}C$ and the maximum value reached after cessation of forcing.

The $\delta^{13}$C of DIC in deep and intermediate ocean boxes shows a lead to the 2.4 Myr eccentricity forcing. A close assessment of the first few peaks in the 2.4 Myr bandpass filtered deep-ocean $\delta^{13}$C reveals that the lead only appears from the second peak onward, indicating that it may be an overshoot in the carbon isotope response to the forcing. Simulations with a transient, linearly increasing forcing were performed to explore whether the proposed overshoot could be found (Fig. 6). In this simulation,

5 the imposed forcing decreased net burial of $C_{org}$ for 500 kyr. During this time, pCO$_2$ progressively increases while $\delta^{13}$C of DIC becomes more depleted. After the forcing has been terminated at $t = 500$ kyr, pCO$_2$ slowly decreases to its initial value over millions of years by the feedback of continental weathering. In contrast, carbon isotopes return to their initial value much more rapidly and subsequently overshoot past it. A maximum $\delta^{13}$C value is reached at $t = 895$ kyr, almost 400 kyr after the forcing has stopped. This confirms that the observed lead of carbon isotopes to the 2.4 Myr eccentricity cycle is an overshoot

10 response rather than an actual lead. Additionally, the slow development of the overshoot explains why it does not result in a lead to the shorter 405 and 100 kyr cycles. The remaining question is why the overshoot develops in deep and intermediate ocean $\delta^{13}$C, but not in the surface ocean (Fig. **??** and Fig. 6). At any time, the mean $\delta^{13}$C value of an ocean box represents a balance between the magnitude of carbon fluxes going in and out of the box and their isotopic signature. The LOSCAR model shows similar results as the box model runs by Kump (1991), which reveal that at steady state, surface ocean $\delta^{13}$C follows the

15 following equation:

$$\delta^{13}\text{C}_{\text{surface}} = \delta^{13}\text{C}_{\text{weathering}} - \Delta^{photo} \times (F_{burial}^{OC}/[F_{burial}^{OC} + F_{burial}^{IC}]) \tag{1}$$

Surface ocean $\delta^{13}$C thus depends on three factors: the ratio between burial rates of organic ($F_{burial}^{OC}$) and inorganic carbon ($F_{burial}^{IC}$), the $\delta^{13}$C of weathered carbon ($\delta^{13}$C$_{\text{weathering}}$), and the magnitude of isotopic fractionation during photosynthesis ($\Delta^{photo}$). As $\delta^{13}$C weathering and $\Delta^{photo}$ are constant parameters in the model, $\delta^{13}$C surface becomes lighter when relatively

20 less organic carbon is buried, and vice versa. The carbon isotopic gradient between surface and deep ocean $\delta^{13}$C can then be calculated according to the following equation (Broecker and Peng, 1982; Tyrrell and Zeebe, 2004):

$$\Delta\delta^{13}\text{C} = -\Delta^{photo} \times \Delta[\text{DIC}]/[\text{DIC}]_{\text{mean}} \tag{2}$$

While $\delta^{13}$C$_{\text{surface}}$ is independent of internal oceanic processes, $\delta^{13}$C$_{\text{deep}}$ is determined by $\Delta^{photo}$ and the ratio between deep and surface DIC ($\Delta[\text{DIC}]$) compared to the mean DIC of the ocean ($[\text{DIC}]_{mean}$). As $\Delta[\text{DIC}]$ is dominantly controlled by the

25 organic carbon pump, this can be interpreted as maintenance of the surface-to-deep isotope gradient by the biological pump.

Since both $\Delta^{photo}$ and the strength of the biological pump are constant in the model, changes in the gradient are mainly caused by the size of the total oceanic carbon inventory, reflected by $[\text{DIC}]_{\text{mean}}$. During the first 500 kyr of the simulation, both $\delta^{13}$C surface and $\delta^{13}$C deep become more $^{13}$C-depleted as $C_{org}$ burial is decreasing relative to IC burial. After the forcing has terminated, $C_{org}$ burial instantaneously increases back to initial conditions. However, $[\text{DIC}]_{\text{mean}}$ and pCO$_2$ are still en-

30 hanced relative to the initial conditions for several Myr, thereby elevating both silicate and carbonate weathering (equations 1 and 2). During this rebound phase, $\delta^{13}$C surface slowly increases back to its initial value, as burial of IC is enhanced relative to $C_{org}$ burial. In addition to this, the elevated inventory of oceanic carbon diminishes the isotope gradient by decreasing

[Figure]

**Figure 7.** Absolute phasing (lags and leads indistinguishable) in absolute ages (kyr, log axis) as a function of frequency (cycles/kyr).

$\Delta[\text{DIC}]/[\text{DIC}]_{\text{mean}}$ (the small increase in $\Delta[\text{DIC}]$ is negligible compared to that in $[\text{DIC}]_{\text{mean}}$). This can be interpreted as a relative decrease in strength of the organic carbon pump compared to the size of the total carbon inventory, causing the deep isotopes to increase even more than those of the surface and resulting in the observed overshoot of $\delta^{13}C_{\text{DP}}$.

**6  2.4 Myr **amplitude modulation of the shorter eccentricity cycles**

[Figure]

**Figure 8.** The envelopes (black) of the 100 and 405 kyr filtered (red) composite record (gray) show a clear 2.4 Myr cycle in $\delta^{13}$C (left side and $\delta^{18}$O (right side. This is additional evidence for 2.4 Myr amplitude modulation of the 100 and 405 kyr eccentricity cycles.

[Figure]

**Figure 9.**  Evolutive multitaper harmonics analysis (c) of the  model output $\delta^{13}C$ (grey) after detrending (black) the <1 Myr signal (red line) in the time (a),  frequency (b) domain. Periods of interest are labelled (b) at the same axis as (c). Distances between peaks in the 100 kyr period of short eccentricity are marked (gray boxes with duration). A window size of 0.7 Myr was used,  $\delta^{13}C$ when forced with a step size of 10 kyr (lines on the  right side in (c) **(1)**. Methods adapted from Pälike et al. (2006).

[Figure]

**Figure 10.** Example time-series model output for an ETP-forced (1:0.5:-0.4) run with `orbstrength` 0.5 and `noiselevel` 0, showing all model output tracers, except PO₄, because it is constant throughout at $2.62\,\mu\mathrm{mol\,kg^{-1}}$. 405 kyr (frequency of $2.46 \pm 0.15\,\mathrm{Myr^{-1}}$) and 2.4 Myr ($0.42 \pm 0.15\,\mathrm{Myr^{-1}}$) bandpass filters are shown for all records (thick darker lines).

[Figure]

**Figure 11.** Time series model output with eccentricity, obliquity, precession, ETP (1:0.5:−0.4), 30°N and 65°N insolation forcing. Thick lines through the tracers represent 405 kyr and 2.4 Myr filters.